# Wake-like skin patterning and neural activity during octopus sleep

Aditi Pophale[1,5], Kazumichi Shimizu[1,5], Tomoyuki Mano[1,5], Teresa L. Iglesias[2], Kerry Martin[1], Makoto Hiroi[1], Keishu Asada[1], Paulette García Andaluz[1], Thi Thu Van Dinh[1], Leenoy Meshulam[3,4] & Sam Reiter[1✉]

While sleeping, many vertebrate groups alternate between at least two sleep stages: rapid eye movement and slow wave sleep[1–4], in part characterized by wake-like and synchronous brain activity, respectively. Here we delineate neural and behavioural correlates of two stages of sleep in octopuses, marine invertebrates that evolutionarily diverged from vertebrates roughly 550 million years ago (ref. 5) and have independently evolved large brains and behavioural sophistication. 'Quiet' sleep in octopuses is rhythmically interrupted by approximately 60-s bouts of pronounced body movements and rapid changes in skin patterning and texture[6]. We show that these bouts are homeostatically regulated, rapidly reversible and come with increased arousal threshold, representing a distinct 'active' sleep stage. Computational analysis of active sleep skin patterning reveals diverse dynamics through a set of patterns conserved across octopuses and strongly resembling those seen while awake. High-density electrophysiological recordings from the central brain reveal that the local field potential (LFP) activity during active sleep resembles that of waking. LFP activity differs across brain regions, with the strongest activity during active sleep seen in the superior frontal and vertical lobes, anatomically connected regions associated with learning and memory function[7–10]. During quiet sleep, these regions are relatively silent but generate LFP oscillations resembling mammalian sleep spindles[11,12] in frequency and duration. The range of similarities with vertebrates indicates that aspects of two-stage sleep in octopuses may represent convergent features of complex cognition.

Vertebrate rapid eye movements (REMs) and slow wave sleep are characterized by a core set of behavioural and electrophysiological correlates, and proposed cognitive functions[13–15] while showing a rich diversity of species-specific features[15]. If the functions ascribed to two-stage sleep are truly general, then one may expect to find neural and behavioural correlates of two-stage sleep widely among animals showing complex cognitive abilities. Octopuses are among the largest brained invertebrates and demonstrate a range of sophisticated behaviours[16], making them ideal for testing the generality of two-stage sleep. Sleeping cephalopods[17] have been observed to undergo rhythmic bouts of body twitches and rapid changes in skin patterning[6,18], mediated by neural control of large populations of skin pigment cells (chromatophores)[19] among other specialized cell types[20]. In octopus, this has been termed 'active sleep' (AS) and is accompanied by an increased arousal threshold, one of several criteria of sleep[15,21]. Expanding on this previous work, we tested whether octopuses possess two stages of sleep behaviour. We then examined neural activity and skin pattern dynamics during sleeping and waking, by developing new methods for behavioural recording and quantification, light-sheet imaging and LFP recordings using Neuropixels probes in these soft bodied animals.

## Behavioural signatures of sleep

During daylight, nocturnal octopuses (*Octopus laqueus*[22]) closed their eyes, adopting a flat resting posture and a uniformly white skin pattern, previously described hallmarks of octopus quiet sleep (QS)[6,17]. Roughly every 60 min, this behaviour was interrupted by roughly 1-minute periods of rapid transitions through a series of skin patterns (Fig. 1a,b and Supplementary Videos 1 and 2), accompanied by pronounced eye and body movements (Fig. 1c,d) and increased breathing rate and arhythmicity (Fig. 1e,f and Extended Data Fig. 1a–c). We quantified patterning behaviour using a convolutional neural network to segment nine animals from 1,743 h of video, tracking changes in the mean brightness of octopus skin (Fig. 1a,b and Extended Data Fig. 2a–c). During the QS separating active bouts, animals generated brief (7.1 ± 0.3 s, $n = 1,163$ events, six animals) and subtle flashes of colouration with a rate that decreased over the time interval between active bouts (Fig. 1g and Extended Data Fig. 2d–f).

The interval between active bouts was dependent on water temperature, with 1-°C increases resulting in roughly 5-minute decreases between bouts (Fig. 1h, linear model, $R^2 = 0.55$, $F$-statistic versus constant model 291, $P$ value of $2.33 \times 10^{-43}$). The rate of active bouts was

[1]Computational Neuroethology Unit, Okinawa Institute of Science and Technology (OIST) Graduate University, Okinawa, Japan. [2]Marine Animal Research Support Team, Okinawa Institute of Science and Technology Graduate University, Okinawa, Japan. [3]Theoretical Sciences Visiting Program, Okinawa Institute of Science and Technology Graduate University, Okinawa, Japan. [4]Computational Neuroscience Center, University of Washington, Seattle, WA, USA. [5]These authors contributed equally: Aditi Pophale, Kazumichi Shimizu, Tomoyuki Mano. ✉e-mail: samuel.reiter@oist.jp

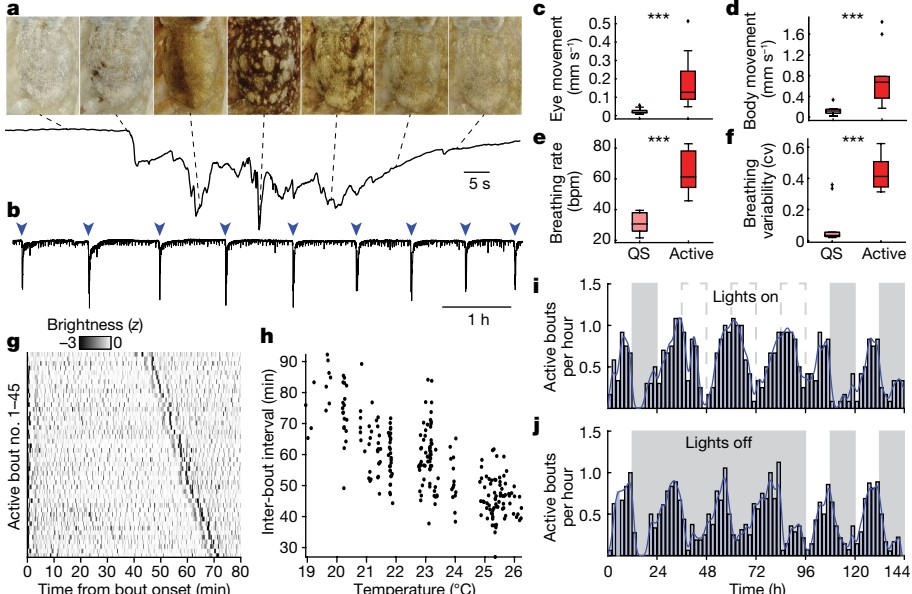

**Fig. 1 | Behavioural correlates of octopus two-stage sleep. a**, Mean skin brightness over time during an active rest bout. The top shows images of octopus body, viewed from the top with head facing up, from throughout the active bout. **b**, Recording mean skin brightness over longer timescales reveals rhythmic alternation between AS and QS. **c–f**, Relative to QS, AS bouts see an increase in eye movements (**c**), body movements (**d**), breathing rate (breaths per minute) (**e**) and breathing variability (coefficient of variation) (**f**). Two-sided Wilcoxon sign rank tests (quiet versus active), $P = 0.00025$, $0.00033$, $0.00018$, $0.00077$, $n = 10$ bouts, three animals. **g**, QS between two active bouts is characterized by repeated flashes of colouration. Rows begin at active bout start, ordered by time to the following active bout ($n = 6$ animals, high-pass filtered $0.005$ Hz for display). **h**, Active bout inter-event interval is temperature dependent ($n = 243$ bouts, ten animals). **i,j**, Circadian rhythm in active bout rate persists over 3 days of constant light (**i**) or darkness (**j**) ($n = 6$ animals, Methods).

strongly modulated over 24 h, peaking during the 12 h of subjective daytime. In a typical 24-h period at 22 °C animals underwent $10 \pm 3.5$ active bouts of $75 \pm 28$ s in duration and $12 \pm 3$ QS bouts of $50.5 \pm 16.43$ min in duration ($n = 3$ animals, mean ± s.d.). This modulation persisted through prolonged periods of constant light or darkness (Fig. 1i,j), suggesting internal control[23] (Rayleigh test, lights on $P = 1.5 \times 10^{-12}$, $n = 322$ bouts, lights off $P = 3.0 \times 10^{-13}$, $n = 318$ bouts). Bout length remained unchanged throughout these manipulations (Extended Data Fig. 2g).

Do active bouts constitute a distinct sleep stage? We first tested arousal levels by delivering mechanical stimulation to animal tanks using a solenoid, and recording animal movement with optical flow (Methods). Animals showed different reactions to mechanical stimulation during QS, active bouts or while awake. Strong and medium (86 and 40 dbV) stimulation produced roughly 1 s of reactionary movements above baseline, regardless of behavioural state, and often resulted in the cessation of pattern dynamics. Weak stimulation (6 dbV) produced movement while awake, but not during QS or active bouts, consistent with results in other cephalopod species (Fig. 2a and Extended Data Fig. 1d). Therefore, active bouts are rapidly reversible states of decreased arousal. Preventing sleep for 2 days (Methods) resulted in a notable increase in the rate of active bouts in the two nights following deprivation (Fig. 2b). Active bouts are therefore homeostatically regulated, meeting another evolutionarily conserved criterion of sleeping behaviour[24]. This regulation was sensitive: specific interruption of an active bout led to the next active bout occurring roughly 22 min sooner than in uninterrupted sleep (Fig. 2c,d and Methods). We therefore refer to two stages of sleep behaviour in octopus: AS and QS.

## Neural activity during AS

To examine neural activity during octopus sleep, we developed techniques for performing electrophysiological recordings from the central brain (supra-oesophageal mass) of head-fixed octopuses using multi-site Neuropixels probes ($n = 9$ animals per probe insertion). To localize recordings we used tissue clearing and light-sheet imaging,

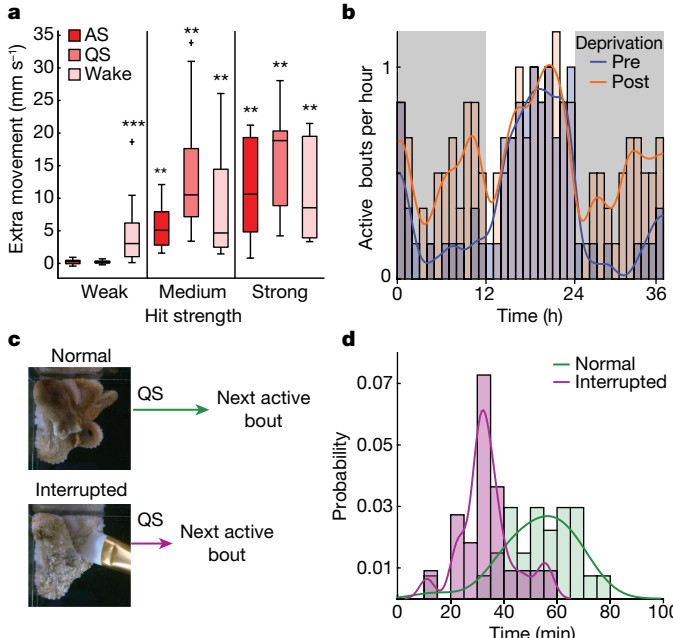

**Fig. 2 | Testing behavioural criteria of sleep. a**, Relative to waking, animals show heightened arousal threshold to mechanical stimulation during QS and AS bouts. Weak (6 dbV), medium (40 dbV) and strong (86 dbV) hit strengths. Two-sided Wilcoxon sign rank tests, $P = 0.19$, $0.27$, $0.0001$, $0.0039$, $0.0039$, $0.0002$, $0.0078$, $0.002$ and $0.002$, $n = 13$, $12$, $21$, $9$, $9$, $13$, $8$, $10$ and $10$ trials (left to right), from $n = 5$ animals. **b**, Increase in active bout rate following 2-day deprivation. Wilcoxon rank sum tests, $P = 0.0065$, $0.0216$ for night 1 and night 2, following deprivation. $n = 15/37$ and $8/31$ bouts (pre-/post-), from six animals. **c**, Schematic of AS bout interruption experiment. **d**, The period of QS separating two active bouts shortens following active bout interruption. Wilcoxon rank sum test, $P = 3.0 \times 10^{-6}$, $n = 22/27$ bouts (normal/interrupted) from three animals.

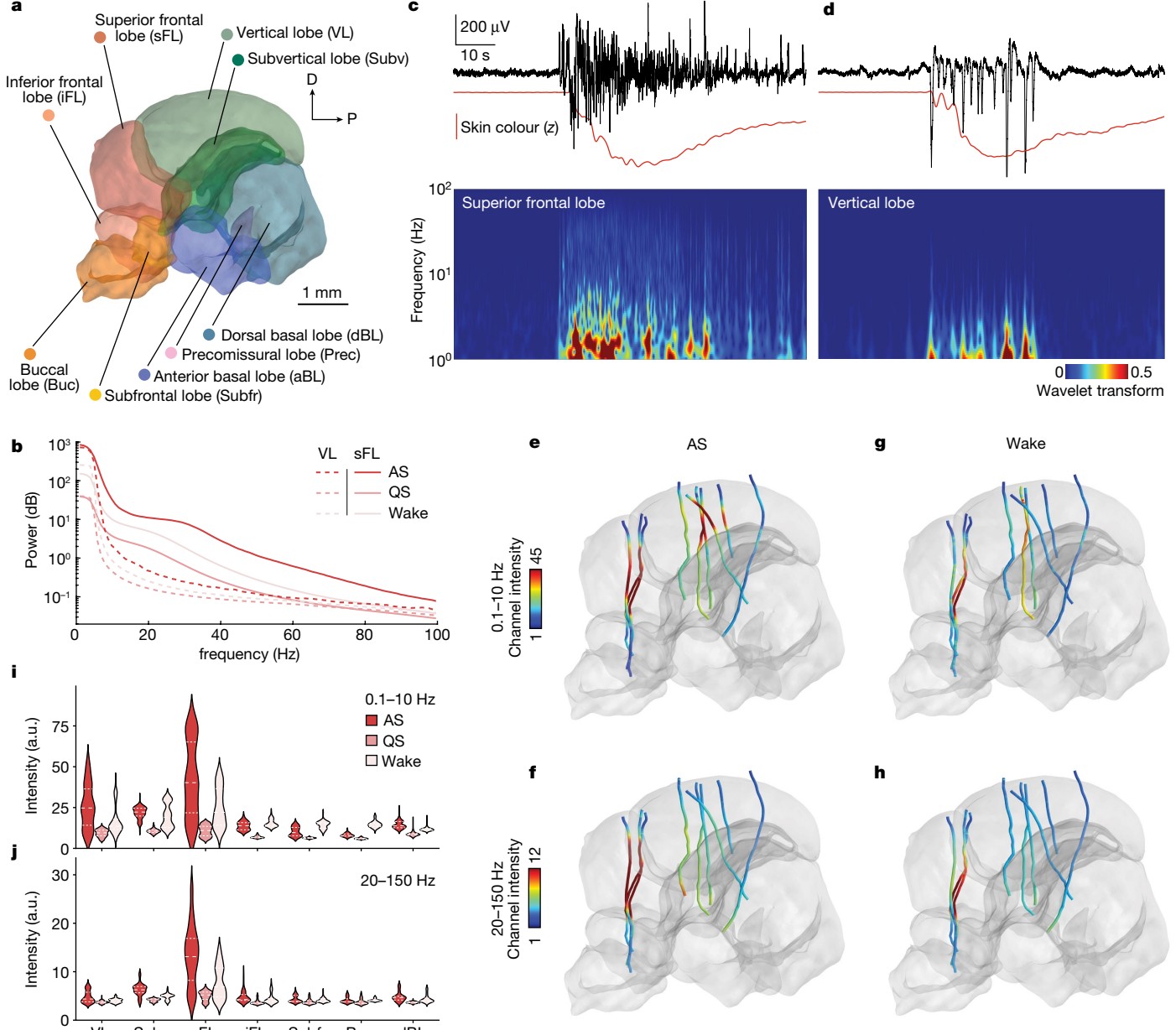

**Fig. 3 | Neural correlates of AS. a**, Atlas of the supra-oesophageal mass, onto which all Neuropixels recordings were mapped. **b**, LFP power spectrum during AS, QS and wake taken from sFL (solid lines) and VL (dashed lines). **c,d**, Representative LFP signals from sFL (**c**) and VL (**d**) at the onset of AS are shown as the top black lines. The red lines underneath represent mean skin brightness, showing the behavioural onset of AS. The bottom shows spectrograms of the corresponding LFP signals (normalized 0–1, Methods)

**e,f**, LFP signal during AS. *n* = 9 Neuropixels recordings were mapped to the atlas. Each probe is coloured with the intensity of low (0.1–10 Hz) (**e**) and high (20–150 Hz) (**f**) frequency oscillations. **g,h**, LFP signal during the wake phase: low, 0.1–10 Hz (**g**) and high, 20–150 Hz (**h**). **i,j**, Violin plots showing the intensity of low- (**i**) and high- (**j**) frequency oscillations during AS, QS and wake phases. All channels from *n* = 9 probes were pooled together.

computationally registering all experiments into a three-dimensional (3D) reference brain atlas that we constructed (Fig. 3a, Extended Data Figs. 3 and 4, Supplementary Video 3 and Methods). In our brain atlas, we manually segmented the central brain (supra-oesophageal mass) into nine large brain regions, following detailed anatomical reports[25,26].

Octopuses fell asleep during neural recordings, showing periods of QS interrupted by rhythmic AS bouts with duration and interval similar to those of AS in freely behaving animals (Extended Data Fig. 5). LFP recordings from the superior frontal lobe (sFL) and vertical lobe (VL), brain regions associated with learning and memory function[7–10], showed levels of LFP activity that differed according to brain state (Fig. 3b).

In both areas, AS was accompanied by large increases in LFP activity over that of QS, with waking activity being of intermediate strength. LFP frequency content differed across regions. The sFL generated activity over a wide frequency band, including prominent 30-Hz oscillations (Fig. 3c and Extended Data Fig. 6b–d). By contrast, the VL reliably produced a series of large (up to approximately 700 μV), low-frequency waveforms (Fig. 3d).

To systematically compare neural activity between AS and waking, we examined LFP strength across brain regions in a low-frequency (0.1–10 Hz) and a high-frequency (20–150 Hz) band (Fig. 2e–j and Methods). In general, there was a strong correlation between a brain

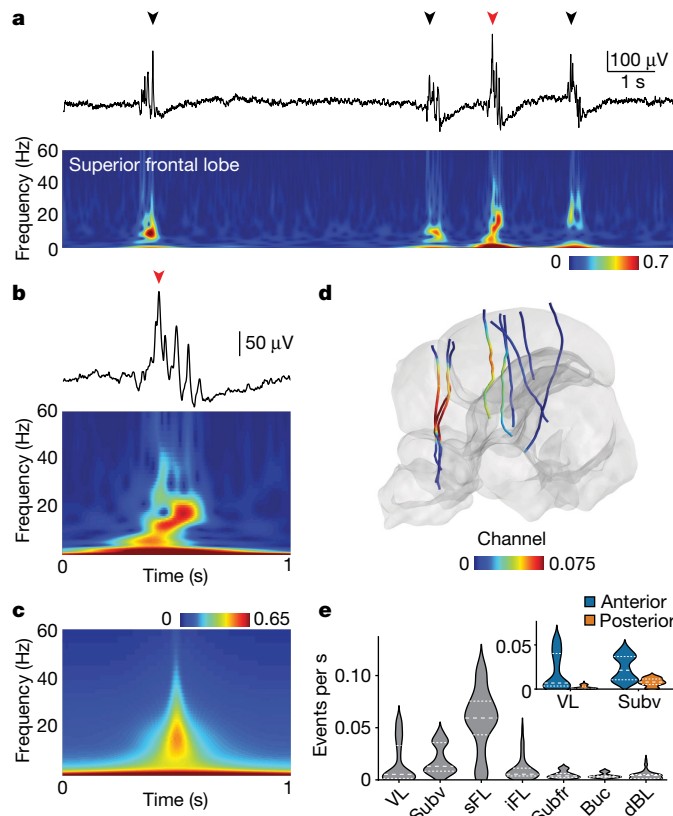

**Fig. 4 | Neural correlates of QS. a**, The top shows the LFP recorded in the sFL during QS, showing oscillatory events (arrow heads) and reduced activity relative to other behavioural states. The bottom shows a spectrogram of top LFP (normalized 0–1, Methods). **b**, Expanded view of burst in (**a**) (red arrow head). **c**, Average spectrogram of oscillatory events (*n* = 3,268, single recording). **d**, Oscillatory events during QS. *n* = 9 Neuropixels recordings were mapped to the atlas. Probe colour relates to the average oscillatory event rate. **e**, Violin plot showing the oscillatory event rate averaged over electrodes in each area. The inset shows that VL and Subv was divided anterior-posteriorly (Methods), showing higher oscillatory event rates anteriorly.

region's activity while awake and during AS (Pearson's *R* = 0.74 and 0.95 for low and high frequency, respectively). Brain regions differed in the relative strength of low- versus high-frequency activity (Fig. 3i,j). The subfrontal lobe (Subfr) and buccal lobe (Buc) showed stronger low-frequency activity during waking than during AS. Other brain regions, in particular the sFL, were more active during AS than waking (Extended Data Fig. 6e,f).

## Neural activity during QS

During QS, the octopus brain was relatively silent, with LFP activity lower than that of AS across frequencies and recorded brain areas (Fig. 3i,j and Extended Data Fig. 7a,b). We found two prominent sources of activity. The first was tied to the brief flashes of skin colouration seen during QS (Fig. 1g and Extended Data Fig. 2d–f). These behavioural events were accompanied by LFP activity resembling that of waking across brain regions (Pearson's *R* = 0.94, 0.99 activity strength correlation for low frequency, high frequency; Extended Data Fig. 7c–e).

We found a second source of activity in the sFL. Here, a nearly silent LFP was punctuated by 12–18-Hz oscillatory events lasting up to 1 s (Fig. 4a–c and Methods). These events were most often not associated with any discernible behavioural change, with only 11% appearing within 10 s of QS colour flashes (2,066 out of 18,058 detected events, *n* = 3 animals). A heavy-tailed inter-event interval distribution suggested a

non-rhythmic generation mechanism (Extended Data Fig. 7f). Searching throughout all recorded brain regions by looking for peaks in the filtered LFP (4–40 Hz, Methods), we found similar events selectively in anterior areas of the VL and subvertical lobe (Subv) (Fig. 4d,e and Extended Data Fig. 7g,h). This hints at functional interactions across parts of the sFL–VL complex during QS, consistent with the direct anatomical connectivity between these regions[8,10].

## AS skin patterning

AS skin patterns operate under direct neural control, thus providing a unique window into the contents of neural activity in the offline brain. To analyse the rapid skin pattern changes observable during AS (Fig. 1a and Supplementary Video 1), we recorded 8K video of 98 AS bouts from three octopuses, filming a top-down view. To extract a robust and expressive quantitative description valid across animals, we used a neural network (Mask R-CNN[27]) to segment the octopus mantle in every frame and used a pretrained VGG-19 neural network to quantify skin patterns as 512-dimensional vectors[28] (Methods). Parallel analysis showed the space of AS skin patterns estimated from our data to be roughly 60-dimensional (59.6 ± 0.3), with patterns of different octopuses largely overlapping (silhouette score 0.0497 ± 1.8 × 10⁻⁴, Extended Data Fig. 8a,c and Methods). Stochasticity in the location of new chromatophore insertion into the skin[29,30] means that at microscopic scales no two octopuses, or even the same octopus on different days, show the same pattern. Here we focus on macroscopic pattern appearance.

We next analysed the trajectories of AS patterns traversing skin pattern space. Between starting and ending with a uniform white skin pattern, AS trajectories traced out diverse and complex paths (Fig. 5a). At a given elapsed time of AS, any two trajectories were on average roughly six times further away from each other (inter-trajectory distance) than they were to the next point in time along a single AS trajectory (intra-trajectory distance, Fig. 5b). Pairs of AS trajectories remained distant even after using dynamic time warping (Extended Data Fig. 8b). AS bouts therefore do not sequence through the same set of skin patterns at different speeds. However, similar patterns appeared at different times across AS bouts. The distribution of nearest patterns between pairs of trajectories, irrespective of time, overlapped with the distribution of intra-trajectory distances. This process is visualized in Fig. 5c: patterns extracted every 10 s from a single AS trajectory (Fig. 5a(i)) showed the characteristic diversity of AS dynamics (Extended Data Fig. 8d). The closest points to these patterns, taken from other AS trajectories of the same octopus as well as from other octopuses, were similar in appearance. In sum, AS trajectory dynamics were diverse, showing a set of patterns without stereotyped sequence, conserved across animals, which at times intersected each other.

While waking, *O. laqueus* can generate a range of skin patterns to camouflage in different natural environments (Extended Data Fig. 9), as well as for social and threat displays. In the laboratory, waking octopuses would occasionally adopt a flat posture in which different skin patterns were shown in full. Analysis of these patterns showed that they fell within the space of skin patterns observed during AS (Extended Data Fig. 8b,c). To look at matching in detail, we identified pairs of images from the same octopus during AS and waking. Non-linear warping from waking to AS patterns revealed a precise alignment in pattern structure between the two (Fig. 5d, Extended Data Fig. 10 and Supplementary Video 4). This suggests that AS dynamics include rapid transitions through skin patterns shown in awake, behaving animals.

## Discussion

Octopuses possess at least two stages of sleep: 'quiet' (QS) and 'active' (AS). Rhythmic AS bouts are homeostatically regulated and robust to temperature and lighting manipulations, indicative of an actively

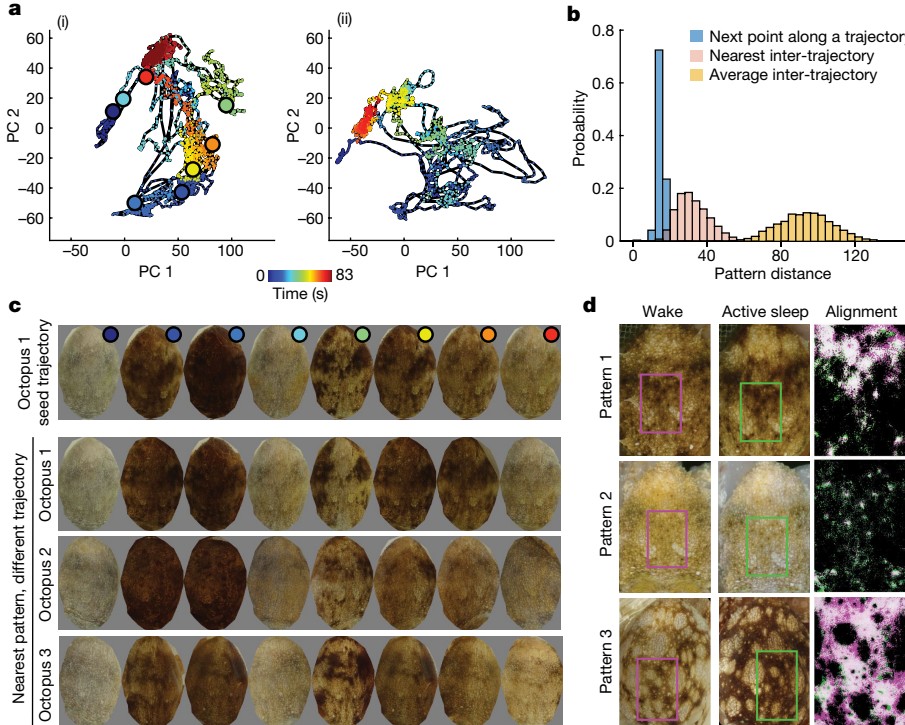

**Fig. 5 | Dynamics of AS skin patterning. a**, Two example AS bout trajectories ((i) and (ii)) projected onto the first two principal components of AS pattern space. Large dots in (i) show points sampled every 10 s from throughout the trajectory. **b**, Histograms showing distributions of pattern distances between (blue) nearest points in time along a trajectory, (pink) nearest points between trajectories, and (yellow) inter-trajectory distance at 0 time lag. Values are averages over AS bouts. **c**, The top row shows octopus 1 skin patterns at 10-s intervals along the trajectory in **a** (i). The bottom rows show nearest skin patterns to each image in the top seed trajectory, for other trajectories of octopus 1 and for other octopuses. **d**, Example pairs of similar waking and sleeping patterns. The right column shows non-linear alignment of rectangular regions in the left and middle columns, with brightness thresholded to show pattern match (white colour, Methods).

maintained biological phenomenon of central importance[31,32]. QS shows 12–18-Hz oscillations in areas associated with learning and memory (the frontal-VL system)[7–10,33], resembling mammalian sleep spindles in frequency and duration[11,12]. However, QS seems to differ from vertebrate slow wave sleep in showing no low-frequency oscillations entraining large areas of the brain[12]. The brief flashes of colouration seen during QS are accompanied by neural activity levels resembling waking, albeit at lower amplitude. Whether this constitutes brief periods of waking, micro-arousal states[34], a kind of QS or a distinct sleep stage remains unclear. AS resembles vertebrate REM sleep in terms of wake-like neural activity accompanying eye and body twitches[15,35]. Coordinated postural changes (for example, arm reaching) are not seen during AS, potentially indicating some level of muscular inhibition. However, a lack of anatomical homology complicates comparison with the atonia of skeletal muscles found in vertebrate REM sleep[35]. Future work will investigate the depth of these similarities mechanistically.

Furthermore, during AS octopuses rapidly transition through sets of skin patterns that strongly resemble those seen while awake. This normally occurs in the safety of the octopus den, and therefore does not broadcast the animal's position to predators. Why do octopuses perform this pronounced sleep behaviour? One possibility is that it represents periods of offline refinement of skin pattern control, analogous to processes thought to occur during vertebrate motor learning[36,37]. Another possibility is that it reflects the reactivation of neural activity underlying waking experience more broadly, reminiscent of vertebrate phenomena linked to memory consolidation such as rodent hippocampal replay[38,39] and the structured activity in the head direction system during mammalian REM sleep[40,41]. Full investigation of the function of AS will require studying whether patterns can be manipulated, as well as a greater understanding of the ethological context in which waking skin patterns are expressed[42–44].

Cephalopod skin patterns seem to be organized hierarchically, with putative higher-order motor control circuitry coordinating large groups of chromatophores to generate macroscopic pattern elements[19,29,45–48]. AS dynamics are consistent with the pseudo-random activation of this high-level control system. It may be possible to infer interactions between motor control elements by studying the statistics of pattern activation. In this way, AS dynamics may also be useful for understanding the logic of waking skin pattern control.

While initially observed in humans[2], recent work has established two-stage sleep across many vertebrate species[1–4]. Our results complement several behavioural reports in cephalopods[6,18] and arthropods[49] of similar active and QS stages. Given the evolutionary distances, these phenomena probably evolved independently from each other, and may represent convergent solutions to shared problems facing complex agents[50]. If such solutions indeed exist, then the high-dimensional and interpretable readout of neural activity in octopus AS skin patterns may help to uncover general principles of two-stage sleep.

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

# Methods

## Experimental animals

All research and animal care procedures were carried out in accordance with institutional guidelines, approved by the OIST Animal Care and Use Committee under approval numbers 2019-244-6 and 2022-364. Adult octopuses (*O. laqueus*, mantle length roughly 3 cm) of both sexes were collected in Okinawan tidal pools and housed in 12 l tanks connected to a seawater system with open circulation to the ocean. Animals were provided with an enriched environment including sand, plants, rocks and coral rubble, as well as a shelter (terracotta pot).

*O. laqueus* were carefully selected for this study after assessing several other options due to (1) their compact brain and body size made them suitable for Neuropixels recording and light-sheet imaging, (2) their white resting skin pattern aided detection of AS bouts, (3) their nocturnal behaviour meant we could film sleep behaviour under white lighting and (4) they were locally available, a regulatory requirement for keeping in the OIST marine station. The brain of *O. laqueus* resembles coastal diurnal octopuses in possessing a seven-gyrus VL (Extended Data Fig. 3d). The VL occupies 9.05% of central brain volume, slightly higher than other coastal nocturnal species, such as the commonly studied *O. vulgaris* and *O. bimaculoides*[51].

## Behavioural filming

Experiments were conducted in closed seawater systems, circulating filtered natural seawater. Water was filtered mechanically and biologically, UV sterilized, oxygenated and exchanged with fresh seawater twice a week. Unless otherwise reported, temperature was cooled to 22 °C, with lighting alternating on a 12/12-h light/dark cycle with a 30-min taper in light intensity. Animals were fed live crabs three times a week during subjective night, while awake. Animals were given a 2-week acclimation period of living in experimental tanks. Experiments started after animals were seen to be resting normally during the daytime.

Low-resolution recordings (Fig. 1b,g–j and Extended Data Figs. 2 and 5b) were filmed using three custom filming chambers. Each chamber placed a single 4K camera (Basler ace acA4024-29uc, 4,024 × 3,036 pixels, 24 fps) viewing four transparent acrylic 100 × 150 × 100 mm tanks from the top, with 13.4 pixel per mm resolution using 12-mm lenses. Lighting was mounted on two sides of the group of four tanks, with white LED day lighting (Koval Smart Aquarium Light, 300 mm, 654 lx) and red LED night lighting (Leimac IDBA-HM300R, 300 × 40 mm Barlight, 129 lx). Other than during feeding, a 5-mm-thick glass cover was placed over the tanks to prevent animal escape. For recording animal movements (Fig. 1c–f and Extended Data Fig. 1), we placed animals in transparent acrylic 300 × 200 × 200 mm tanks fitted with shelters (three-dimensionally printed and terracotta pots), with shelter entrance facing the tank wall. Cameras (as above, 24 or 30 fps) were positioned facing the shelters of single animals. Lights (Leimac IDBA-HM300W, 300 × 40 mm, 3 klx) were placed on tank top and/or sides.

Recordings were made with PylonRecorder2 (v.0.6), using online hardware compression to h264 format and writing to solid-state drives. A single computer running Windows 10 was equipped with two graphics cards (Nvidia Quadro P1000 and Quadro P5000), which run up to seven cameras simultaneously.

High-resolution recordings (Figs. 1a and 5) were made by placing an 8K camera (Canon EOS R5, 8,192 × 5,464 pixels, 30 fps) fitted with a Canon Macro Lens EF 180 mm lens on a gimbal, and filming top-down on a single animal using a 45° mirror (96 pixels per mm). Three white bar lights (Leimac IDBA-HM300W, 300 × 40 mm, 3,933 lx) were used to light the tank. All recordings were made using the following camera settings: 1/200, F32, ISO 3,200. Because AS events are periodic, AS recording was started a few minutes before the expected AS time, and ended after AS pattern dynamics ceased. Some pairs of waking and sleeping patterns were shot at 4K resolution (Basler ace acA4112-30uc, 4,096 × 3,000 pixels, 30 fps, 32.3 pixels per mm, 1,000 lx), looking down on animals using a 50-mm lens. High-resolution recordings were shot at 24 °C (room temperature water).

To put lighting levels in context, daylight ranges from 10 to 25 klx when sunny, 1,000 lx when overcast. Nighttime light levels range from roughly 0.3 lx during a full moon to 0.002 lx when the moon is not visible. Animal sleep time, AS duration and interval seemed normal under various experimental lighting conditions (Extended Data Fig. 5). Darker skin patterns did not appear under 3 klx lighting while awake, but were observed under 1 klx lighting.

## Behavioural experiments

For measuring arousal threshold, mechanical stimulation was delivered using a solenoid, fixed in a constant position on the tank wall and controlled by an Arduino (Arduino Mega 2560). Synchronization of camera and solenoid hit time was done using a red LED placed in view of the camera and out of view of the animal. Stimulus strength was calibrated using a hydrophone (DolphinEar DE200), placed in the position of the octopus. Three strengths of stimulus 6, 40 and 86 dbV were delivered to the octopus during active bouts, QS and wake in the terracotta pot. QS was defined as the interval between two active bouts, in which the octopus had the characteristic lack of movement, flat posture, smooth texture, closed eyes and white colour. All trials were conducted between 12:00 and 17:00.

For sleep homeostasis, octopuses were recorded continuously for 48 h before the start of sleep deprivation. The following day, animals were kept awake from 07:00 to 17:00 by gently brushing their skin with a paintbrush every 2–3 min. Movement, elevated posture and eye opening were considered indicators of wakefulness. Animals were left to behave freely after 17:00. Sleep deprivation was then repeated, using the above procedure, for a second day. Post-deprivation behaviour was subsequently recorded for 48 h without any disturbance. For the active bout interruption experiment, a random subset of active bouts were interrupted using a paintbrush. The amount of QS preceding the following AS bout was then compared in interrupted and uninterrupted 'trials'.

For measuring circadian rhythm, octopuses were acclimated to a 12-h light/dark cycle for 2 weeks. After 48 h of continuous filming, they were subject to either 72 h of continuous daylight or 72 h of continuous darkness. Feeding was halted during this period to prevent cues from live crabs. Animals were then switched back to a 12-h light/dark cycle.

For measuring temperature dependence, the temperature of the water circulating in the behavioural filming system was cooled to 19, 22 and 24 °C using a cooler (Poafamx AL-300) attached to the water circulation system. Each temperature was maintained for 2 days. Higher temperatures were achieved by turning off the cooler. Tank water temperature was measured using a temperature sensor (Tinytag Aquatic 2 TG-4100).

## Surgery for electrophysiology

Animals were anaesthetized (2% ethanol in filtered sea water) more than 3 days before the recordings, and the distal 2–4 cm of all arms were surgically shortened to prevent them from removing future head fixation. Wounds were sealed with tissue glue (Histoacryl, B.Braun). Following this procedure, animals recovered in an experimental tank equipped with a closed seawater circulation system (above), in which they could move and eat immediately on waking from anaesthesia. Octopus did not demonstrate signs of pain (that is, arm grooming behaviour[52]), therefore local analgesia was not applied, preventing administration related stress and sleep disruption. One day before recording, animals were anaesthetized as above and the top of the head was placed out of the water. The skin, soft tissue and muscle over the cartilaginous head capsule were removed with microscissors. A three-dimensionally printed plastic ring with a pole was glued to the head capsule over the central brain with tissue glue and dental light cure adhesive (3M, Transbond XT Light Cure Paste Adhesive). The inside of the ring was filled with silicone sealant (Kwik-Cast, WPI) to prevent the

wound from touching sea water. Animals were left to recover overnight in the recording tank. On the day of recording, animals were anaesthetized as above. A small hole was cut into the head capsule to expose the central brain, and the sheath surrounding the brain was removed using fine forceps. The ring was covered again using silicone sealant. Fresh seawater was washed over the animal, before moving it to the experimental tank. It was then head-fixed using a metal rod. Animals recovered from anaesthesia within minutes of exposure to fresh seawater. The water level in the recording tank was reduced to have the top of the animal's head above the water line during probe insertion. The animal's body was supported with an aquarium wool filter mat. A Neuropixels v.1.0 probe shank was coated with CM-DiI (100 μg to 50 μl ethanol, Thermo Fisher Scientific, CellTracker CM-DiI Dye) for post hoc probe position localization. The probe was mounted on a motorized micromanipulator (New Scale Technologies, M3-LS). After removing the cured silicone sealant, the Neuropixels probe was lowered into the brain at the speed of 200 μm min$^{-1}$. The depth of the recordings varied across experiments. We explored a range of depths, insertion sites and angles. After lowering, the inside of the head-fixation plastic ring was again filled with silicone sealant and water level was raised so that the animal's head was submerged. The recording tank circulated aerated seawater at a rate of roughly 0.2 l min$^{-1}$ at room temperature (24 °C).

## Electrophysiological recording

Neuropixels recordings were performed using SpikeGLX software (v.3.0). with sampling rates at 2.5 kHz (for LFP signals) and 30 kHz (for extracellular spike signals). The mantle of the animal was simultaneously filmed using a 4K camera as in behavioural recordings (Behavioural filming above, shot at 1,034 lx). Video was synchronized to electrophysiological recording by sending a 25-Hz transistor-transistor-logic (TTL) signal from an Arduino to trigger camera frame exposure and to log TTL time using spikeGLX. Animals showed AS 12.7 ± 6.6 (s.d.) h after recordings began, demonstrating AS bouts similar in duration and interval to that of freely behaving animals (Extended Data Fig. 5).

## Tissue clearing and light-sheet imaging

Following recordings, the Neuropixels probe was removed from the brain and the animal was euthanized by gradually increasing ethanol concentration from 2 to 5% in sea water. The head of the recorded animal was dissected, with tissues surrounding the head capsule removed as much as possible. The brain in the head capsule was fixed in 4% paraformaldehyde at 4 °C for 24–48 h. Dissected brain tissue was cleared using a second-generation CUBIC method[53]. First, the tissue was incubated in 50% CUBIC-L solution at 25 °C overnight, followed by 100% CUBIC-L incubation at 25 °C for 24 h. After PBS wash, the tissue was immersed in BOBO-1 nuclear staining dye (ThemoFisher B3582; 1/800 dilution) for 3 days. The tissue was washed with PBS, then placed in 50% CUBIC-R overnight followed by 100% CUBIC-R for 24 h (Extended Data Fig. 3a). The cleared sample was embedded in a transparent agarose gel for mounting on a microscope. To scan the brain, we custom-built a light-sheet microscope using the technique of axially swept light-sheet microscopy[54]. The microscope was equipped with a 10X detection objective lens (Olympus XLPLN10XSVMP) and a 10X illumination objective lens (KYOCERA SOC Corporation, CS03-10-30-152). Images were acquired with $(x, y, z) = (0.65, 0.65, 2.5)$ μm resolution. BOBO-1 was imaged with a 488 nm excitation laser and 536/40 nm bandpass filter, whereas CM-DiI was imaged with a 532-nm excitation laser and 593/40-nm bandpass filter (Extended Data Fig. 3b,c).

## Brain registration and atlas construction

To construct a reference brain atlas of *O. laqueus*, we cleared and stained an adult octopus brain following the procedure described above and scanned the brain with $(x, y, z) = (0.65, 0.65, 2.5)$ μm resolution. We cropped the supra-oesophageal mass from the whole brain image and downsampled it to $(x, y, z) = (10, 10, 10)$ μm. We used 3D

Slicer[55] to manually annotate this 3D image, referencing existing anatomical atlases[25,26] (Extended Data Fig. 3d,e). 3D images of individual octopus brains were mapped to this reference atlas using the symmetric image normalization method (SyN) method implemented in the Advanced Normalization Tools (ANTs) library[56]. Before the registration, the BOBO-1 channel was downsampled to $(x, y, z) = (10, 10, 10)$ μm. A binary mask was then created manually using 3D Slicer, masking tissues other than the supra-oesophageal mass. The brain was then mapped to the reference brain by a two-step transformation. First, an affine transformation was computed to roughly align the two brains, using mutual information as a metric function. Second, a non-linear warping was computed using the SyN algorithm with cross-correlation as a metric function. In ANTs command line, the following parameters were used: --transform Affine[0.1] --metric MI[${fix_img},${mov_img},1,128,Regular,0.5] --convergence [1000x1000x1000,1e-5,15] --shrink-factors 8x4x2 --smoothing-sigmas 3x2x1vox --transform SyN[0.1,4.0,0.2] --metric CC[${fix_img},${mov_img},1,6] --convergence [500x500x100x30,1e-6,10] --shrink-factors 8x4x2x1 --smoothing-sigmas 3x2x1x0vox. This generated an affine transformation matrix and a warp field given as a four-dimensional matrix (Extended Data Fig. 3g). We controlled the parameters in ANTs to prevent excessive warping, which was quantified by the values of Jacobian determinants (Extended Data Fig. 3g). The atlas and the registered brain were overlaid, showing visually precise alignments (Extended Data Fig. 3f). Alignment quality was quantified by computing the voxel-wise normalized cross-correlation[56] value with window radius of 4 voxels (Extended Data Fig. 3h), which showed positive values in most of the areas and especially high positive values at the boundary between brain lobes. We also generated an average nuclear stained image from $n = 9$ independently aligned brains (Extended Data Fig. 3h). The lobe structure was maintained, further supporting the quality of our registration.

To analyse the location of a Neuropixels probe, the CM-DiI channel from a 3D brain image was first downsampled to $(x, y, z) = (5, 5, 5)$ μm. The CM-DiI probe track was then manually labelled using 3D Slicer. This labelled track image was smoothed by first skeletonizing the binary object using the morphology.skeletonize_3d function implemented in the scikit-image library, and then fitting the resulting skeleton with a B-spline. The $xyz$ coordinates of Neuropixels probe channels were then mapped to the reference space using the transformation computed above. Finally, each recording channel was assigned a unique region ID on the basis of the atlas region. Determining probe depth from CM-DiI images is sometimes a non-trivial problem due to the spread of the dye by diffusion. Following previous heuristic treatments[57], after mapping we inspected the characteristic LFP patterns at the boundary of the anatomical regions. If necessary, we shifted the probe location along the depth axis to increase LFP-anatomy correspondence. The automatically determined locations were usually very accurate, and the maximum correction was ten channels (roughly 100 μm).

To divide brain regions into anterior and posterior halves (Fig. 4e inset), we took the most anterior point and posterior point of each brain region as its A–P minimum and maximum values and computed the midpoint between them as (min + max)/2. Channels were divided into anterior or posterior on the basis of whether they were anterior or posterior to the brain region midpoint.

## Behavioural analysis

To measure octopus skin brightness in behavioural recordings (Fig. 1), we segmented octopuses from background with the FAIR Detectron2 platform[58] (v.0.1.3), using a pretrained base model (COCO Instance Segmentation with Mask R-CNN, R50-FPN, 3× schedule), fine-tuned with octopus training datasets. Training set labelling was done using the Labelbox platform. To accelerate data processing, octopuses were segmented every 100 frames (4.16 s), with mean intensity calculated on every frame using the nearest preceding segmentation. Multi-day

videos were processed in parallel by using FFMPEG to cut videos into 1-h clips, running each clip independently, then recombining.

Colour flashes during QS were detected using the mean skin brightness trace during QS. The trace was filtered 0.005–2 Hz using a three-pole Butterworth filter, and peaks were detected on the negative of the $z$-scored signal, with a minimum peak height of 2, minimum prominence of 0.5 and minimum separation in time of 10 s. Colour flash duration was calculated by taking a window of the mean skin brightness trace from 500 frames (21 s) before a colour flash peak to 1,000 frames (42 s) after a peak. This time series was $z$-scored, and threshold crossings (2$z$) on either side of the peak time were taken as the start and end time.

For sleep time, duration and interval, AS and QS times were identified manually using the mean skin brightness recording coupled with video confirmation. Wake times were similarly identified manually. AS inter-event intervals were calculated between bouts in which the animal did not wake up. AS duration was determined by considering a window 10 s before and 100 s after AS start times. These time series were $z$-scored, and low pass filtered at 0.1 Hz using a two-pole Butterworth filter. The length of the largest continuous stretch of data falling below a threshold was taken as the AS duration. We explored a range of thresholds (Extended Data Fig. 5d), deciding on 0.2 as a good subjective match to video data.

Histograms in Fig. 1i,j used 2-h binning, and a continuous rate estimate was calculated by smoothing a 0.1-h binning using a Gaussian kernel with s.d. 40 bins (4 h). Figure 2b was calculated similarly, using 1-h histogram binning. Figure 2d used 5-min binning, a probability density was estimated using a kernel density estimate (MATLAB 'ksdensity').

For movement analysis, in the arousal threshold experiments (Fig. 2a and Extended Data Fig. 1g–i), we extracted clips 1 s before stimulation time to 1 s after. Animals were segmented from background (as above) on the first frame of a clip. Within the segmentation mask, prominent features were detected using SIFT[59] keypoint detection (contrast threshold of 0.05). Lukas–Kanade optical flow[60] (window size of 512 pixels) was then used to track these points over frames. Movement magnitude was calculated as the mean optic flow magnitude between neighbouring frames. For calculating reactionary movements, a baseline mean magnitude for 25 frames (1 s) before stimulation time was subtracted from the mean magnitude for 25 frames (1 s) following stimulation.

For analysis of animal movements during QS and AS (Fig. 1c–f and Extended Data Fig. 1a–c), we extracted 2-min clips centred on AS start times. The eye, body and the anterior mantle (for measuring breathing) were manually segmented from the first frame of this clip. Movement magnitude was calculated as above, separately for each segmentation mask. To isolate eye and breathing movements from overall body movements, the average movement within the eye/anterior mantle mask was removed from each frame and we have reported residual movements. Breathing rate was extracted from anterior mantle residual movements, with inhalation detected through peak detection in the $z$-scored, smoothed (10 frames) trace, with a peak prominence of 0.05. Breathing rate was then linearly interpolated to video frame rate. Figure 1 reports average movement magnitude for the first 30 s of the clip (QS), and the third 30 s of the clip (AS). Calculation of breathing arythmicity for waking animals (Extended Data Fig. 1b) was calculated as QS/AS, on separate 30-s video clips.

Behavioural analysis was performed using OIST's Saion HPC system, using up to 32 GPUs (Nvidia V100 and P100s). Core analysis was written in Python (v.3.6 and 3.7), with further analysis written using MATLAB 2019a.

### Electrophysiological analysis
LFP data were preprocessed by resampling from 2.5 to 1 kHz, filtering 0.1–150 Hz and re-referencing by subtracting the median of ten channels located out of the brain from all channels. Spectrograms were calculated using a continuous wavelet transform with a Mortlet wavelet

(MATLAB 'cmor1.5-1'), scales logarithmically spaced between 1 and 100 Hz. Spectrograms were normalized in amplitude by dividing all values by the maximum value. Spectra were calculated on non-overlapping 1-s chunks of data using the Chronux toolbox (v.2.12, http://chronux.org/)[61] with a time-bandwidth product of five and nine tapers. The results were then averaged over data chunks.

For calculating channel intensity during different behavioural states, a uniform procedure was conducted on different selections of data. For AS, QS and wakefulness, 60 s of LFP data were loaded, beginning at the transition of every detected AS or wake phase and taking the 60 s before AS times as QS. For QS colour flashes, 700-ms chunks of data were loaded, centred on colouration flash events (detected as above). After data loading, two filtered versions of the data were then generated, at 0.1–10 and 20–150 Hz for every recording channel. The envelope of these filtered signals was calculated using a 150-tap Hilbert filter. Signal strength for a channel was calculated as the mean of this envelope. We median filtered more than five channels to remove noisy channel readings. We then averaged this vector over all events.

For QS oscillation events, 1,200 s of LFP data were loaded preceding every AS bout and any wake events (manually detected, above) were removed. QS oscillation events were detected by filtering the data 4–40 Hz, then finding peaks in the $z$-scored signal with a minimum height of two, minimum prominence of two and a minimum separation of 1 s. Estimates of oscillation event rates were taken per data chunk and averaged. This was then smoothed with a five-channel median filter, as in activity strength measurements. To calculate LFP activity strength over brain regions, we averaged the channel intensity for all electrodes located within a brain region. We required data from a minimum of two probes to consider activity strength for a brain region. Correlations between different activity strength measurements were taken over all electrodes from these brain regions. Unless otherwise stated, filtering was performed using three-pole Butterworth filters.

### Skin pattern analysis
To quantify AS skin patterns, we adapted techniques developed for describing cuttlefish camouflage[28]. High-resolution octopus videos were processed by first detecting the octopus mantle, using the Detectron2 platform[58] as above. Mantles were aligned by choosing a single source image and mapping all images onto this source image by ellipse fitting and similarity transformation. Determination of anterior versus posterior direction was done manually for waking images and the first frame of AS video clips. Images were then cropped and downsampled to 1,004 × 675 pixels (20% image size after cropping to segmented mantle), with background pixels coloured uniform grey (as in Fig. 4c). Following standard preprocessing (zero-centring), 400 × 400 pixel crops of the dorsal mantle were evaluated by a VGG-19 (ref. 62) network pretrained on the ImageNet[63] database, using the Keras[64] platform (included in TensorFlow v.2.0). We used the max-pooled fifth layer activations ('block5_pool') as our skin pattern metric[65]. This resulted in 512-dimensional vectors describing skin patterns for every frame in a video clip.

The starting points of AS trajectory dynamics were aligned across video clips by calculating the first principal component of the 512-feature by frame matrix, and thresholding the absolute value of its approximate derivative (difference between neighbouring points in time, threshold 0.1). All further analysis was done on start-time aligned trajectories. To estimate the dimensionality of AS space, we ran Parallel Analysis[66] several times on 10,000 randomly selected images from the total dataset. To estimate the overlap of different animals' patterns within AS space, we similarly calculated the Silhouette score[67] several times on 10,000 randomly selected images from the total dataset. Intra-trajectory distance was calculated as the mean Euclidean distance between neighbouring points in time along a trajectory. Inter-trajectory distance was calculated between two trajectories as the mean element-wise Euclidean distance, from trajectory start time

to the end of the shorter trajectory. Distances between trajectories after dynamic time warping[68] were divided by trajectory distance to compare with non-time warped inter-trajectory distances. Nearest inter-trajectory distances (between AS bouts and between waking images and AS bouts) were calculated by taking, for all pairs of trajectories, the minimum Euclidean distance between points.

For aligning select waking and AS skin patterns, similar patterns were extracted from videos manually. Precise alignment was achieved by manually selecting corresponding points and interpolating between these points using a moving least-squares algorithm[69] to produce a mapping from the AS image to the waking image. Images were uniformly and linearly brightened for display. Green rectangles in AS images (Fig. 5d and Extended Data Fig. 10) are approximate crops (non-linear mapping). To show overlaid matches, images were grey scaled, inverted and thresholded (image specific threshold of 8-bit grayscale at 180–230) to show the dark pattern regions.

## Statistics and reproducibility

Unless stated otherwise, data are mean ± s.e.m. For box plots, margins are 25th and 75th percentiles; middle line, median; whiskers, boundaries before outliers; outliers (+) are values beyond 1.5× interquartile range from the box margins. Experiments were repeated independently several times with similar results, with numbers of repetitions and sex (female or male) as follows: temperature modulation $n = 9$ animals, arousal threshold $n = 5$ animals, homeostasis $n = 6$ animals, active bout movement $n = 3$ animals, continuous light on/off $n = 6$ animals, electrophysiology $n = 9$ animals, skin pattern dynamics $n = 3$ animals and wake–sleep pattern matching $n = 5$ animals.

## Reporting summary

Further information on research design is available in the Nature Portfolio Reporting Summary linked to this article.

## Data availability

Data are available from the corresponding author on request. A small dataset is provided with the analysis code for demonstration purposes.

## Code availability

The code developed in this study is posted in a repository on GitHub: https://github.com/oist/pophale2023.

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

**Acknowledgements** We are grateful to the OIST Cephalopod Research Support Team for animal care, for the help and support provided by the Scientific Computing and Data Analysis and Engineering sections, Research Support Division at OIST. We thank all Reiter laboratory members for assistance and discussion. We thank G. Laurent, K. Doya, H. Obenhaus, H. Norimoto and S. Shi for comments on the manuscript. This research was supported by the Okinawa Institute of Science and Technology, and Japan Society for the Promotion of Science (JSPS) Kakenhi grant nos. 60869155 and 20K15939. T.M. was financed by a Grant-in-Aid for JSPS Fellows (grant no. JSPS KAKENHI 21J01369). L.M. was financed by the Burroughs Wellcome Fund CASI award, the Swartz Foundation and the OIST theoretical visiting scholar programme.

**Author contributions** The project was defined by A.P., K.S., T.L.I. and S.R. Mechanical design and assembly was done by K.M., M.H. and A.P. Animal care was the responsibility of T.L.I., K.A., A.P. and K.S. Behavioural experiments were carried out by A.P. and T.L.I. Electrophysiological experiments were carried out by K.S. and S.R. Anatomical experiments and analysis were carried out by T.M., P.G.A. and T.T.V.D. Behavioural and electrophysiological analysis were carried out by S.R., A.P. and L.M. S.R. wrote the manuscript with participation of all authors, and supervised the project.

**Competing interests** The authors declare no competing interests.

**Additional information**
**Correspondence and requests for materials** should be addressed to Sam Reiter.

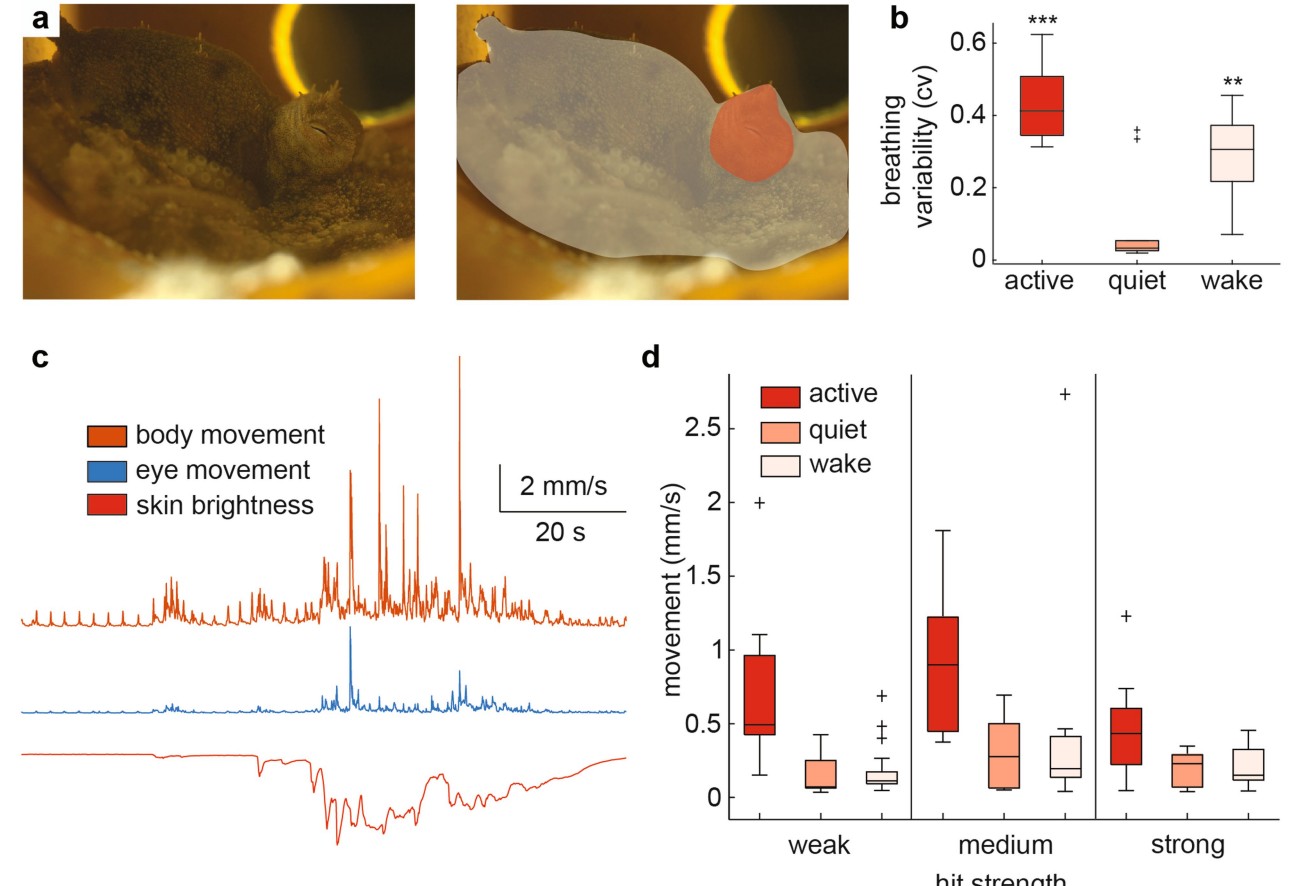

**Extended Data Fig. 1 | Movements during active bouts. a**) Example image of an octopus during active sleep, and manual segmentation of body regions for movement tracking. **b**) QS shows decreased variability in breathing rate (coefficient of variation) compared to active bouts/wake. Two-sided Wilcoxon rank sum tests, p = 0.00077 (QS-active), 0.0076 (QS-wake), N = 10,10,9 bouts (active, QS, wake) from 3 animals. **c**) Time course of increased movements (optic flow magnitude, Methods) during an active bout, identified by changes in skin brightness. **d**) Baseline movements preceding arousal threshold experiments (1s before hit time). N = 13, 12, 21, 9, 9, 13, 8, 10, 10 trials (left to right), from N = 5 animals.

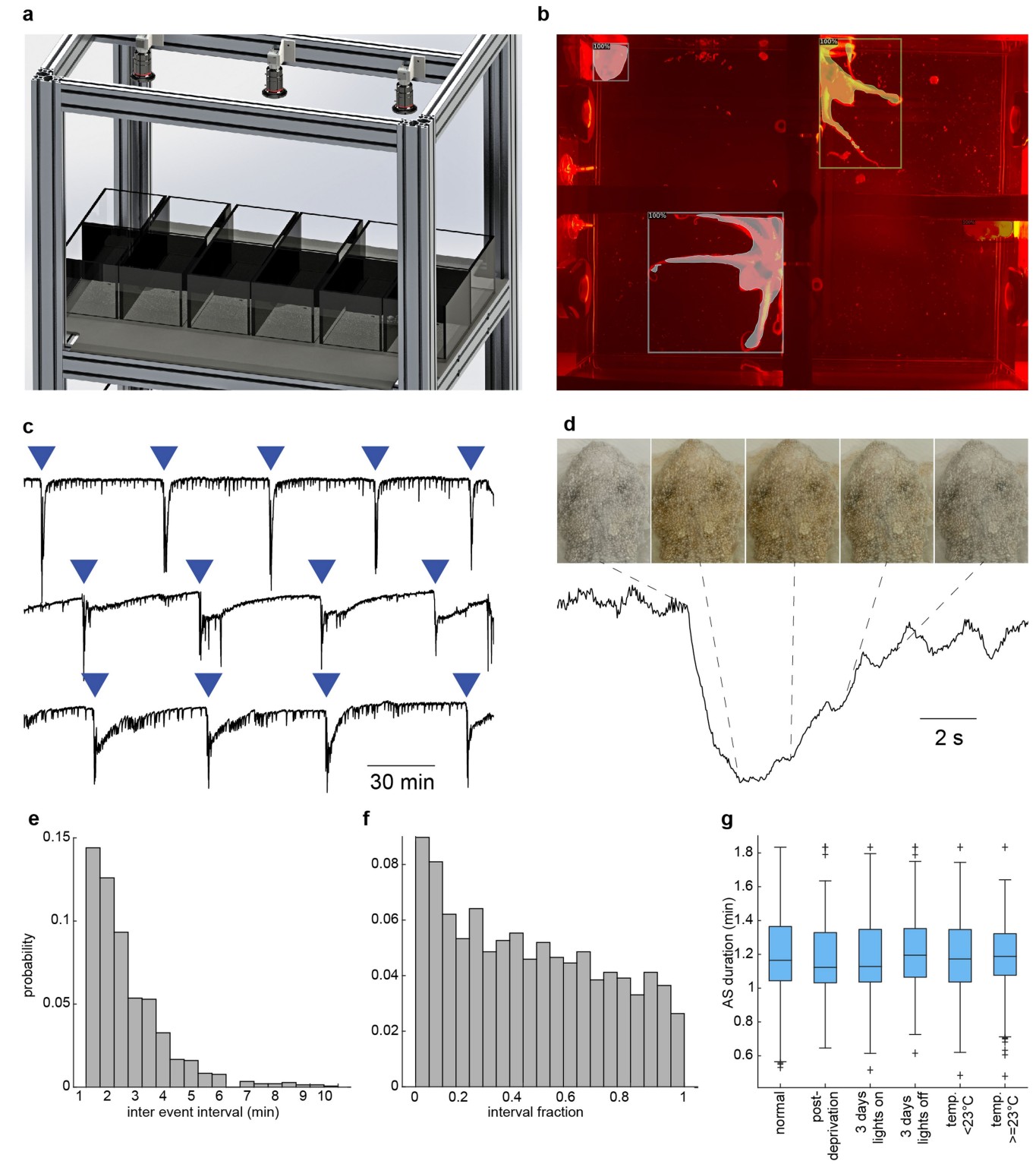

**Extended Data Fig. 2 | Coloration changes during sleep. a**) Rendering of experimental filming setup. **b**) Example image, taken at night under red lighting, with octopuses segmented using a Mask R-CNN. 100% refers to the network's confidence of correct identification. **c**) Time series of mean skin brightness of three octopuses, simultaneously recorded and automatically segmented using a Mask R-CNN. Blue arrowheads: active rest bouts (manually detected, Methods). **d**) An example flash of coloration during QS, recorded at high resolution. Top: example images throughout the event. Bottom: mean skin brightness. **e**) QS colour flash inter-event interval. 3/1437 intervals omitted for display. **f**) QS colour flash occurrence rate decreases as a function of the fraction of time to the next active bout (linear regression R^2 = 0.77, F = 61.8, p = 0, N = 20 histogram bins from 1482 events, 6 animals). **g**) Active bout duration remains constant through manipulations other than decreasing the temperature. N = 528, 131, 316, 317, 164, 178 bouts from N = 6, 6, 6, 6, 10, 10 animals.

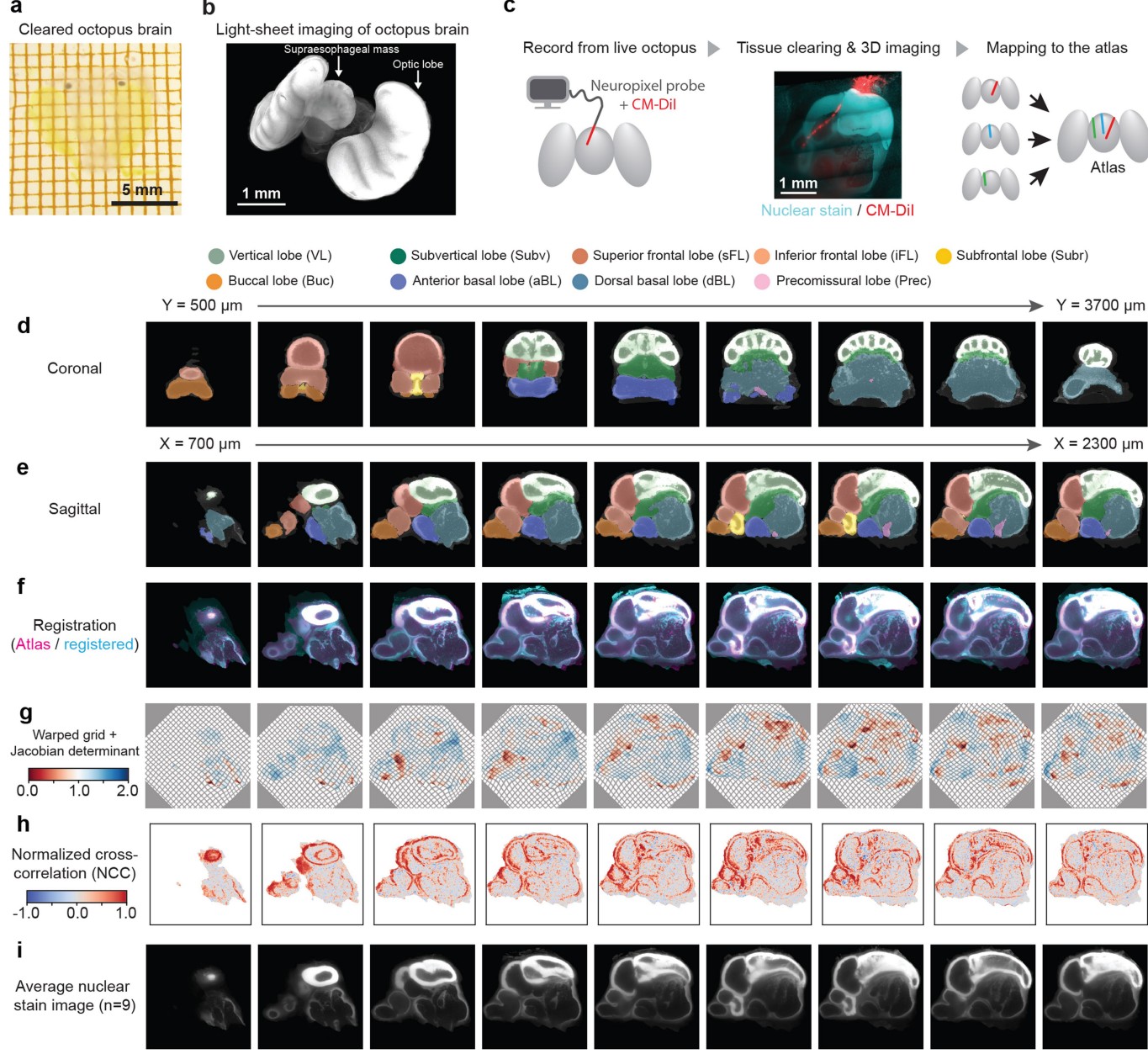

**Extended Data Fig. 3 | Octopus brain atlas and Neuropixels mapping.**
**a)** Adult *O. laqueus* brain, cleared with CUBIC (Methods). **b)** 3D rendering of the cleared octopus brain imaged with a light sheet microscope. **c)** Neuropixels mapping workflow. Neuropixels probe was coated with CM-DiI to leave fluorescent labelling of penetration track in the brain. The brain was cleared and imaged using a light sheet microscope with dual channels (nuclear staining and CM-DiI). Using the nuclear staining channel, we computed the mapping to atlas space. **d)** Coronal sections of *O. laqueus* brain atlas. **e)** Sagittal sections of *O. laqueus* brain atlas. **f)** Representative result of brain registration, where atlas (magenta) and a registered brain (cyan) are overlaid. **g)** Representative warp field generated by registration, overlaid with corresponding Jacobian determinant. **h)** Voxel-wise normalised cross-correlation map between the atlas nuclear staining image and registered brain. (Methods) **i)** An average nuclear staining image generated by N = 9 brains independently mapped to the atlas.

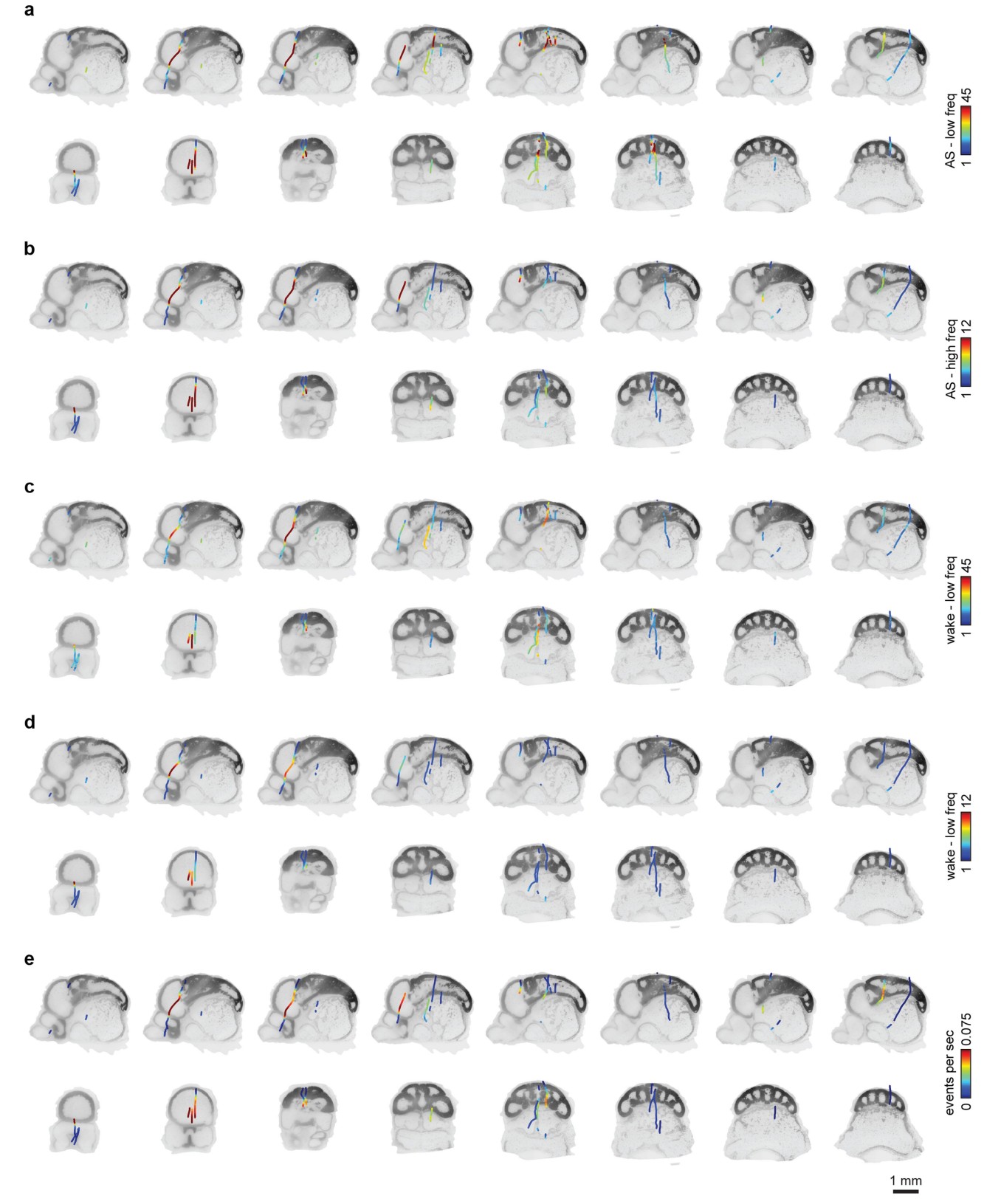

**Extended Data Fig. 4 | Visualisation of Neuropixels probes after brain registration. a,b)** Sagittal (top) and coronal (bottom) slices through 3D reference brain volume, showing mapped Neuropixels probe locations (Methods). Probes are coloured by low (0.1 - 10 Hz, a) and high (20 - 40 Hz, b) frequency oscillation of LFP signal during AS. **c,d)** Same plot as in a) and b) for wake. **e)** Oscillatory bursts during QS.

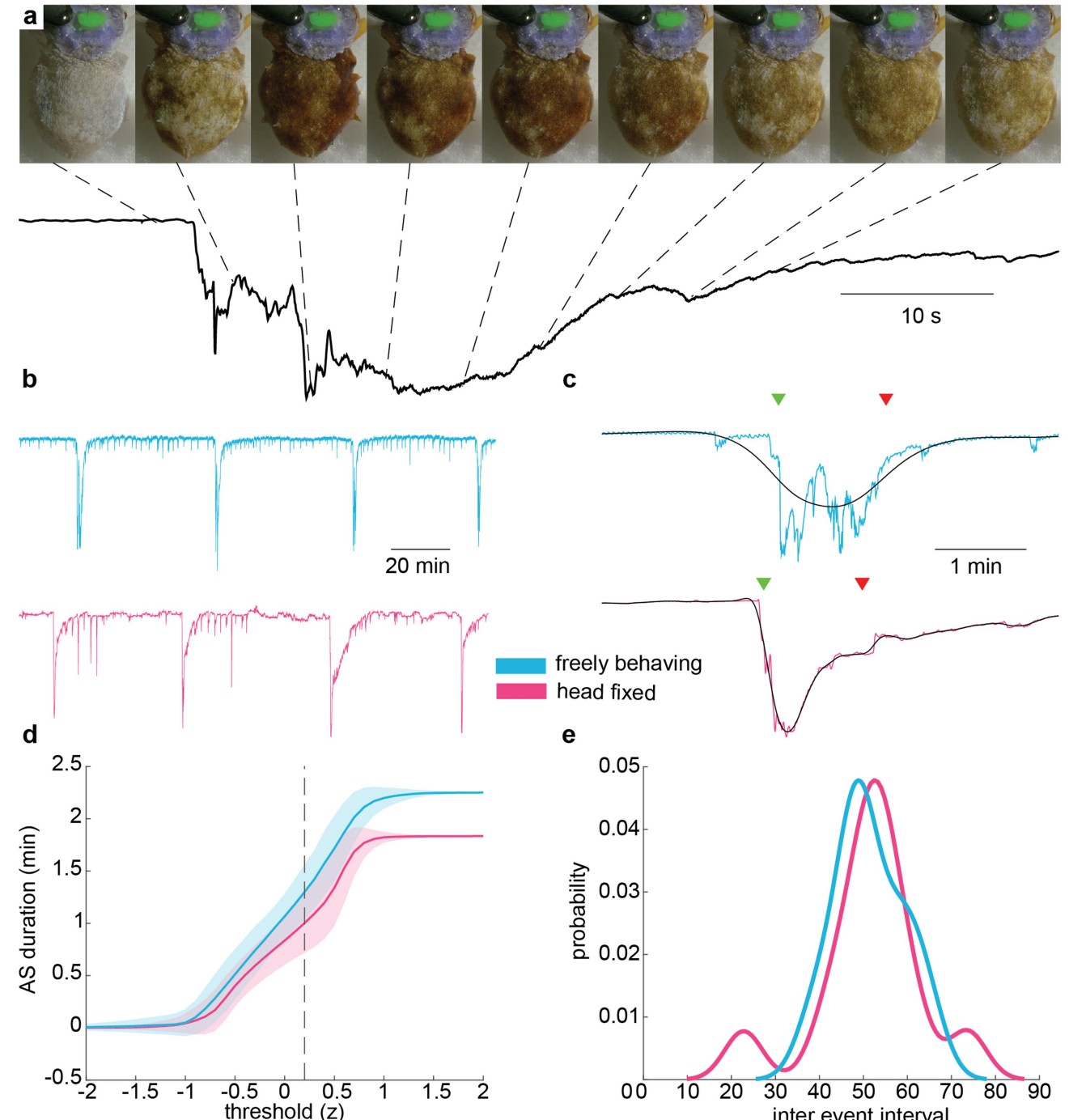

**Extended Data Fig. 5 | Head fixed vs freely moving active sleep. a**) Top: Images of an octopus taken throughout the AS bout (top-down view, images taken 5 s apart). Bottom: mean skin brightness over time during the bout. **b**) Mean skin brightness over time shows QS punctuated rhythmically by AS bouts in freely behaving animals (top) and during head fixation (bottom). **c**) Zoomed in view of single AS bouts, showing filtered data used for calculating AS duration (black), start and end times (green and red arrowheads) for freely behaving and head fixed animals. Qualitative differences between experimental conditions possibly reflect different levels of sleep depth, or recording differences (whole body vs mantle). **d**) AS bout duration is similar in head fixed (N = 76 bouts from 9 animals) and freely behaving (N = 478 bouts from 6 animals) conditions under a range of detection thresholds (Methods). Error band: ±1 SD. **e**) Kernel density estimates of AS bout inter-event intervals are similar in head fixed and freely behaving animals (Normal distribution kernel, freely behaving animal data (N = 14) temperature matched to head fixed data (N = 12), >23.5 °C and <24.5 °C.

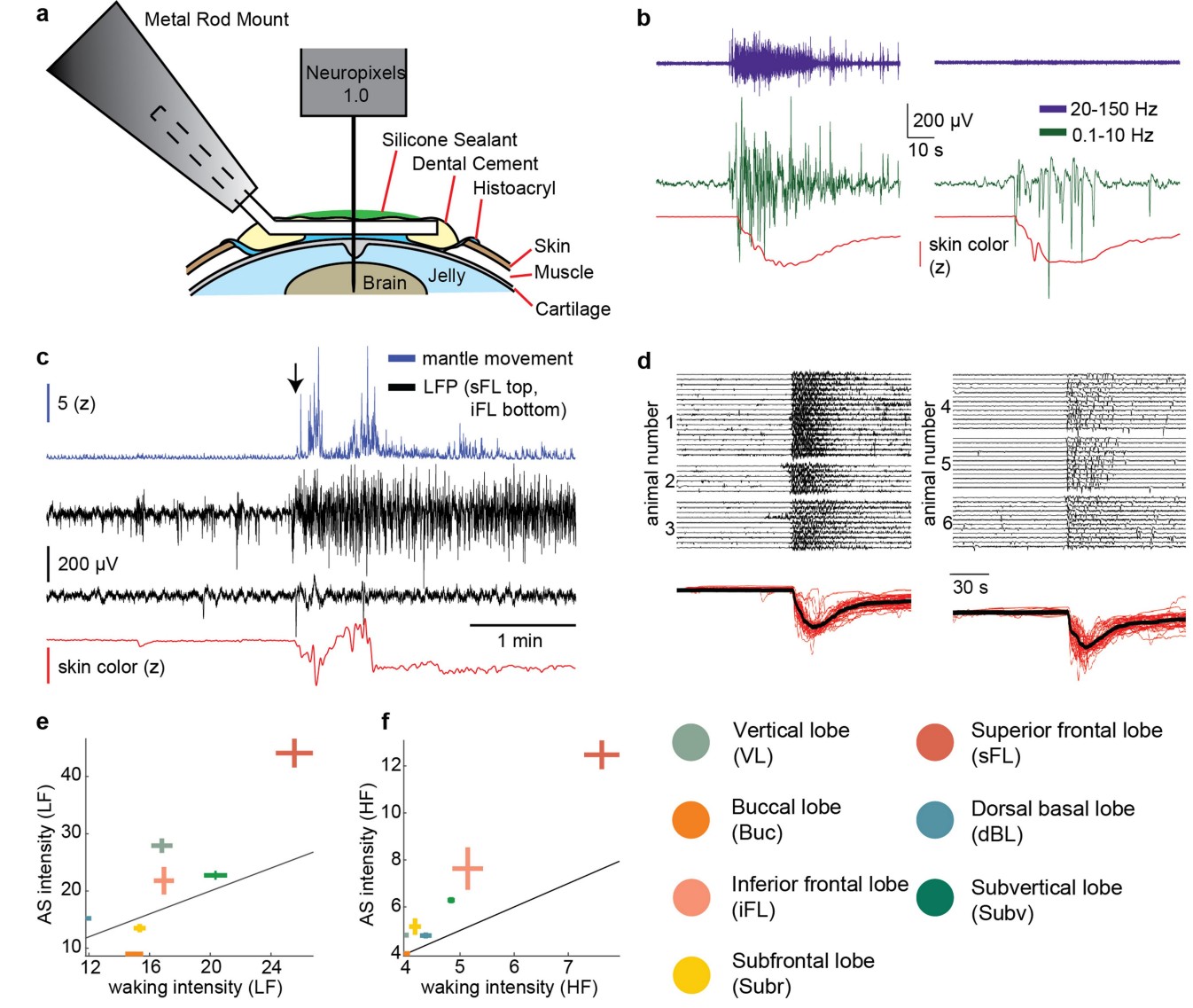

**Extended Data Fig. 6 | Neural correlates of active sleep. a)** Schematic of head fixation technique. **b)** LFP recordings from the sFL (left) and VL (right), as in Fig. 3b,c, filtered for low-frequency (LF) (0.1-10 Hz) and high-frequency (HF) (20-150 Hz) activity. **c)** Time around a sleep-wake transition (black arrow) demonstrating recording stability. Neural activity in the sFL (black, top) increases and mantle coloration (red) darkens upon waking. Activity in the iFL (black, bottom), remains quiet. There are two periods of transient large movements, which are not prominent in either LFP recording. **d)** LFP centred on AS start time (rows: different AS bouts), showing reliability in AS related LFP activity from sFL (left) and VL (right) across animals. **e)** Relationship between low-frequency (0.1-10 Hz) LFP activity strength during waking and AS. Crosses: mean ± 95% confidence interval for all electrodes located in a brain region. Line: Y = X. N = 583, 477, 85, 81, 84, 239, 395 electrodes from N = 8, 3, 3, 2, 3, 3, 6 animals for VL, sFL, iFL, Buc, Subr, dBL, Subv respectfully. **f)** As e) but for high-frequency (20-150 Hz) LFP activity strength.

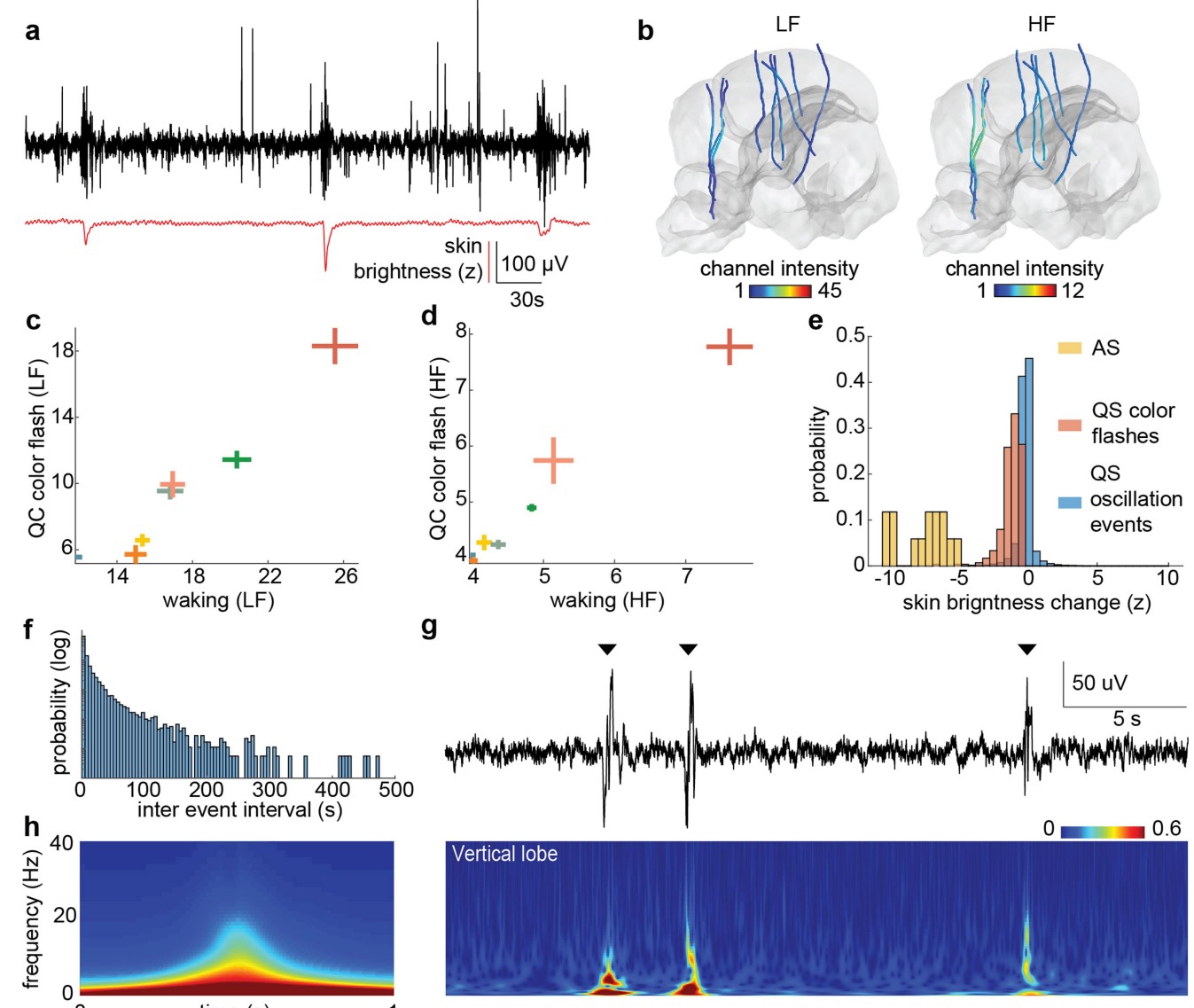

**Extended Data Fig. 7 | Neural correlates of quiet sleep. a**) Example LFP recording from the sFL and skin brightness trace during QS, showing increases in neural activity at times of QS colour flashes. **b**) low-frequency (LF) (0.1-10 Hz) and high-frequency (HF) (20-150 Hz) activity across recording electrodes during QS. Colour scales as in Fig. 3e–h. **c**) Relationship between low-frequency LFP activity strength during waking and QS colour flashes. Crosses correspond to the mean ± 95% confidence interval for all electrodes located in a brain region. Colours denote brain regions, as in Extended Data Fig. 6. N = 583, 477, 85, 81, 84, 239, 395 electrodes from N = 8, 3, 3, 2, 3, 3, 6 animals for VL, sFL, iFL,

Buc, Subr, dBL, Subv respectfully. **d**) As d) but for high-frequency LFP activity strength. **e**) Distributions of the magnitude of skin brightness change for AS bouts, QS colour flashes, and QS oscillation events. **f**) QS oscillation event inter-event interval distribution. **g**) Top: Example LFP recording (filtered 0.5-150 Hz for display) from the anterior VL during QS, showing detected QS oscillatory events (arrowheads). Bottom: Spectrogram of above VL LFP recording (normalised 0-1, Methods). **h**) Average spectrogram over QS oscillatory events detected in the anterior VL (N = 2111, single recording, colour scale as in h).

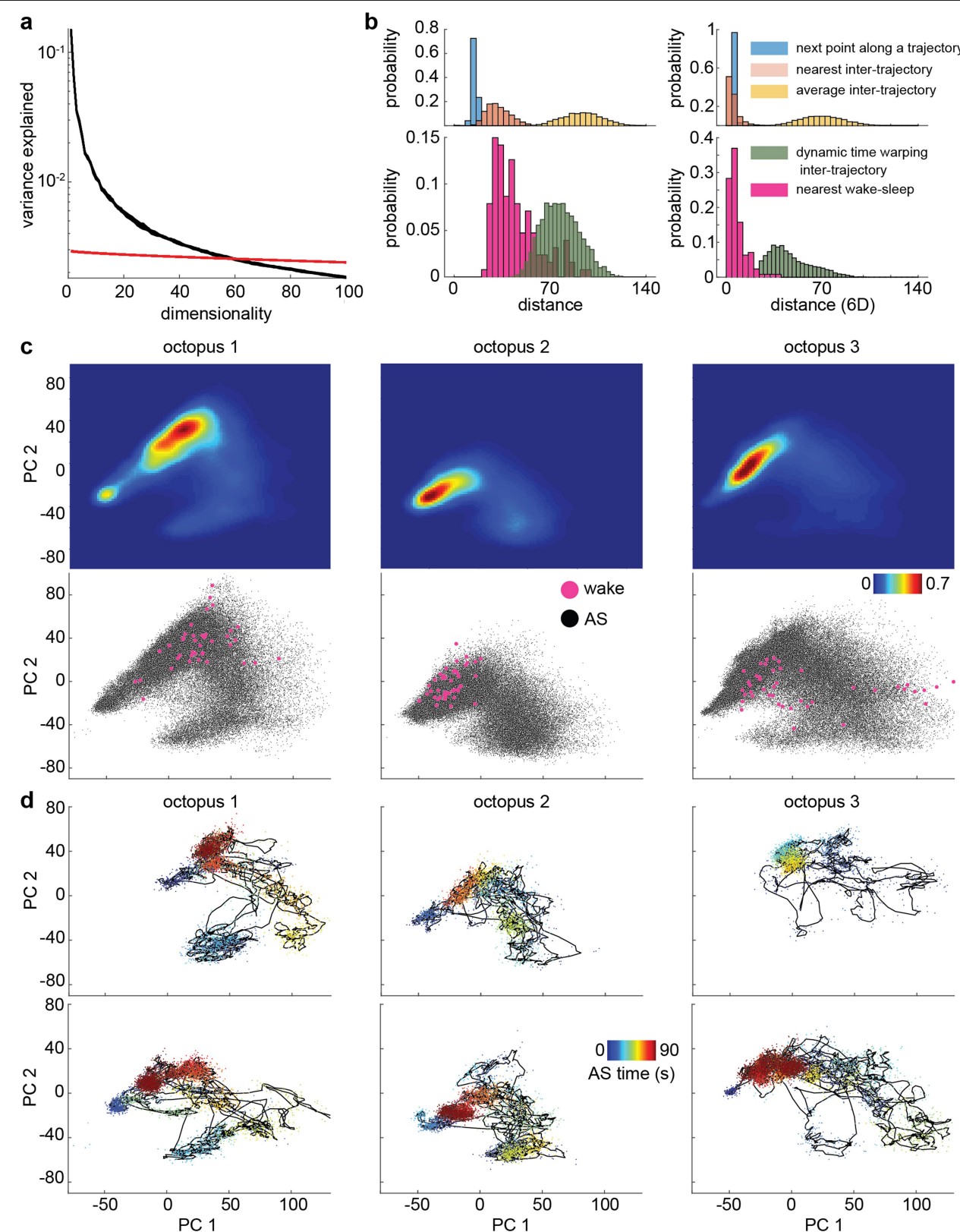

**Extended Data Fig. 8 | Dynamical landscape of skin pattern space. a)** Black traces: Variance explained as a function of principal component dimension for 10 random draws of 10,000 samples. (58±6% of variability explained at 60 dimensions, 285,986 images from 3 animals). Red traces: As black, after independent shuffling of features. **b)** Left Bottom: (Green) Average inter-trajectory distance distribution after dynamic time warping. (Magenta) Distribution of distances between waking patterns and the closest AS pattern. Top: Distance distributions as in Fig. 4b. Right: Similar results as left but for data projected onto top 6 PCs. Silhouette score: 0.076±3.148. **c)** Top: Histogram of occupancy within the top two principal components of AS pattern space, separated by octopus. Occupancy was normalised to the peak occupancy bin (Gaussian smoothing, sd = 2 bins). Bottom: Projection of AS (black) and waking (magenta) points onto the first two principal components. **d)** Two example AS trajectories from each of three animals, projected onto the first two principal components.

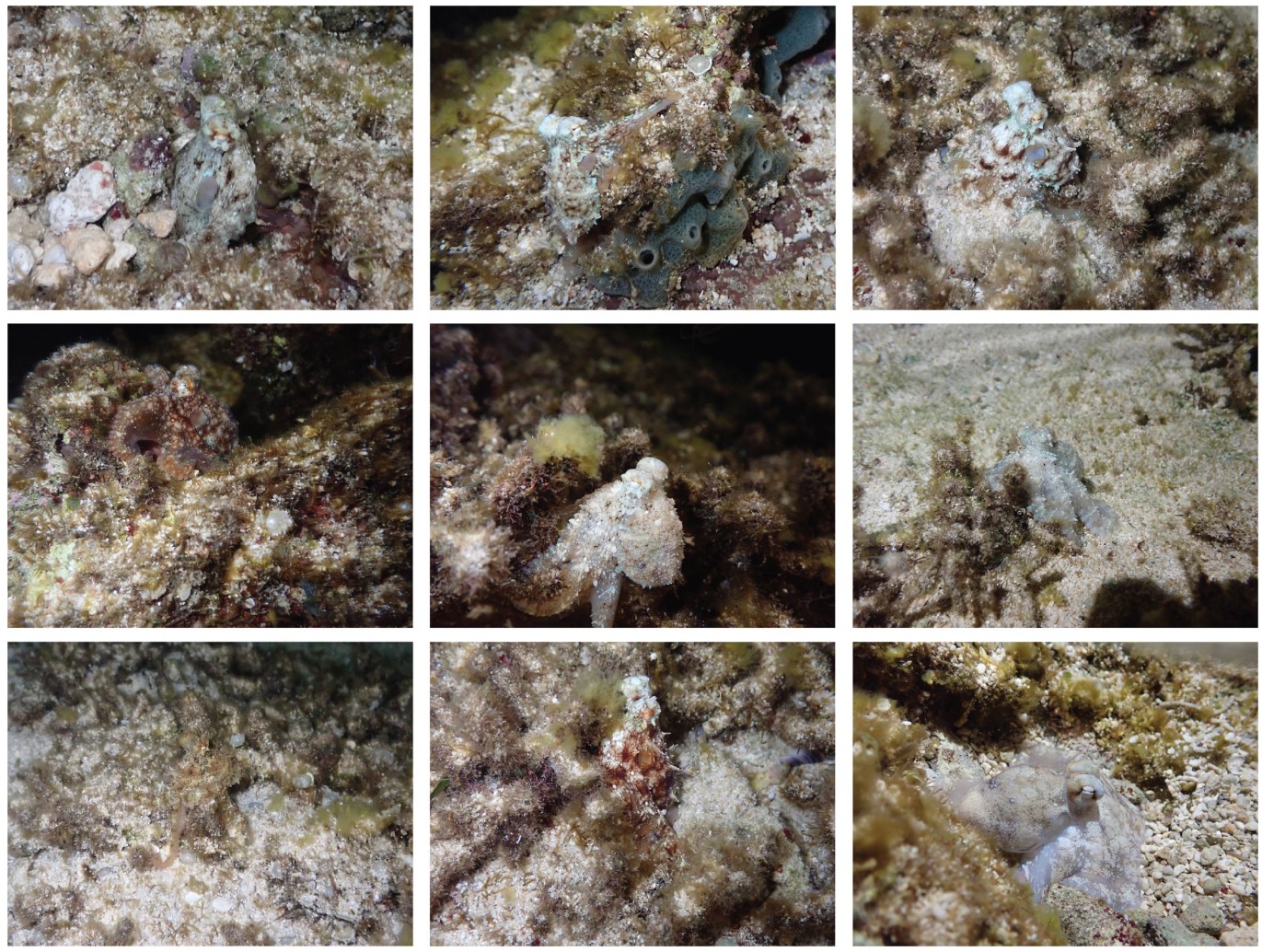

**Extended Data Fig. 9 | O. laqueus patterns in nature.** A collection of images of octopuses adopting different waking skin patterns in different background environments. Top left image shows an animal peeking out of its den, where it sleeps during the day.

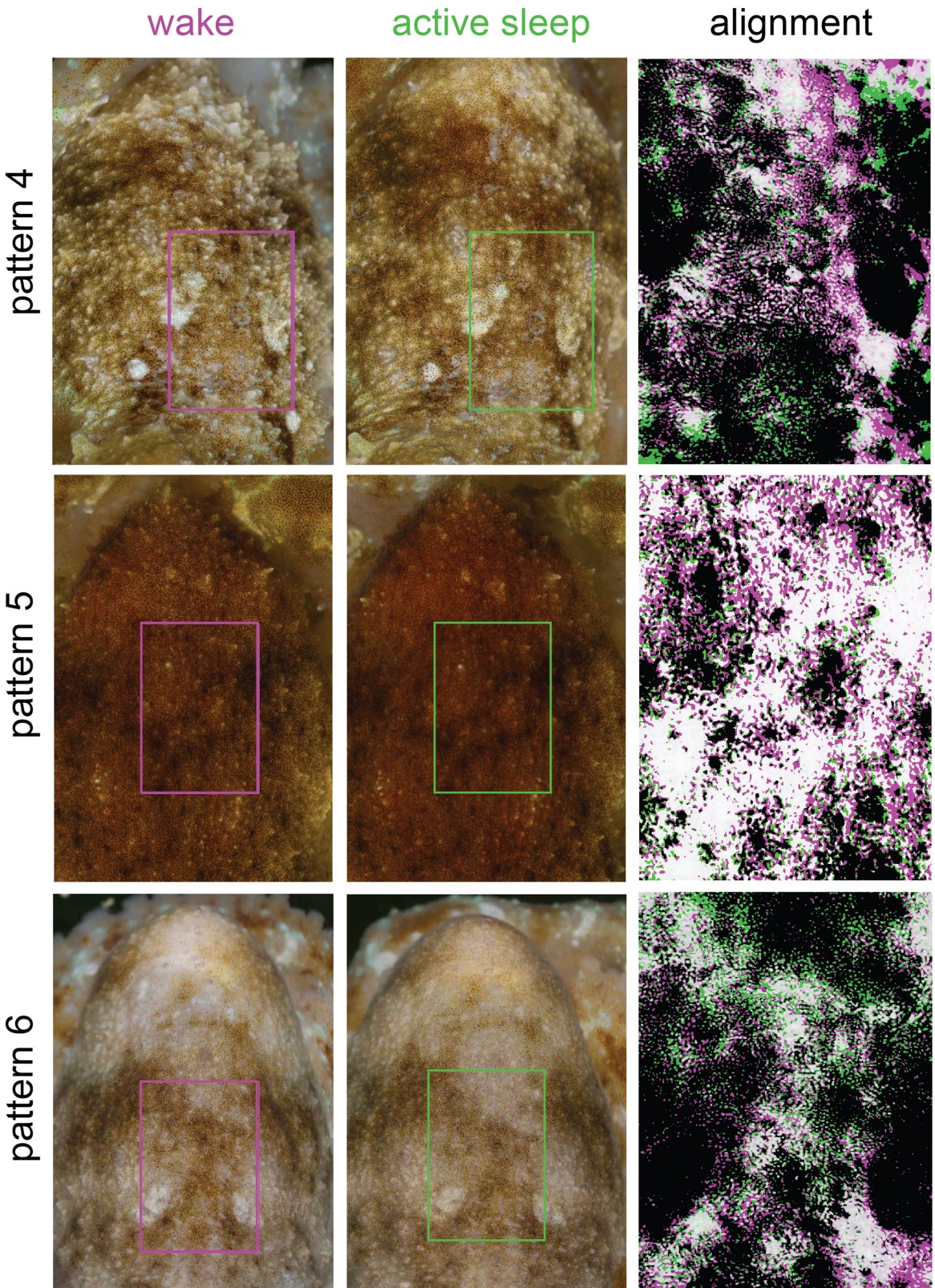

**Extended Data Fig. 10 | Similar patterns during wake and AS.** Further example pairs of similar waking and sleeping patterns (see Fig. 4d). Right column shows nonlinear alignment of rectangular regions in left and middle columns, with brightness thresholded to display pattern match (white colour, Methods).

# Reporting Summary

## Statistics

For all statistical analyses, confirm that the following items are present in the figure legend, table legend, main text, or Methods section.

| n/a | Confirmed | |
|---|---|---|
| ☐ | ☒ | The exact sample size (*n*) for each experimental group/condition, given as a discrete number and unit of measurement |
| ☐ | ☒ | A statement on whether measurements were taken from distinct samples or whether the same sample was measured repeatedly |
| ☐ | ☒ | The statistical test(s) used AND whether they are one- or two-sided<br>*Only common tests should be described solely by name; describe more complex techniques in the Methods section.* |
| ☐ | ☒ | A description of all covariates tested |
| ☐ | ☒ | A description of any assumptions or corrections, such as tests of normality and adjustment for multiple comparisons |
| ☐ | ☒ | A full description of the statistical parameters including central tendency (e.g. means) or other basic estimates (e.g. regression coefficient) AND variation (e.g. standard deviation) or associated estimates of uncertainty (e.g. confidence intervals) |
| ☐ | ☒ | For null hypothesis testing, the test statistic (e.g. *F*, *t*, *r*) with confidence intervals, effect sizes, degrees of freedom and *P* value noted<br>*Give P values as exact values whenever suitable.* |
| ☒ | ☐ | For Bayesian analysis, information on the choice of priors and Markov chain Monte Carlo settings |
| ☒ | ☐ | For hierarchical and complex designs, identification of the appropriate level for tests and full reporting of outcomes |
| ☒ | ☐ | Estimates of effect sizes (e.g. Cohen's *d*, Pearson's *r*), indicating how they were calculated |

*Our web collection on statistics for biologists contains articles on many of the points above.*

## Software and code

Policy information about availability of computer code

| | |
|---|---|
| Data collection | Video data was collected using PylonRecorder2:0.6, a C++ application making use of the Basler Pylon5 SDK. Electrophysiological data was collected using SpikeGLX 3.0., with camera synchronization achieved using an Arduino to activate camera frame exposure and to register TTL times. Light sheet microscopy was performed using custom software written in LabVIEW 2019. |
| Data analysis | Video clips were cut using FFMPEG:3.4.11. Core video analysis was performed using scripts written in Python:3.6 and 3.7. Electrophysiological analysis and further processing of video data was done with Matlab:2019a. Spectra were calculated using the Chronux toolbox:2.12. Segmentation was done using the Detectron2 platform:0.1.3. Neural network analysis was done using Keras, included in tensorflow 2.0. Microscopy data was analyzed using ImageJ:1.53, Imaris:9.7.1, and Slicer:4.1.1 |

For manuscripts utilizing custom algorithms or software that are central to the research but not yet described in published literature, software must be made available to editors and reviewers. We strongly encourage code deposition in a community repository (e.g. GitHub). See the Nature Portfolio guidelines for submitting code & software for further information.

## Data

Policy information about availability of data

All manuscripts must include a data availability statement. This statement should provide the following information, where applicable:

- Accession codes, unique identifiers, or web links for publicly available datasets
- A description of any restrictions on data availability
- For clinical datasets or third party data, please ensure that the statement adheres to our policy

> Data are available from the corresponding author on request. A small dataset is provided with the analysis code for demonstration purposes. Data is available at https://github.com/oist/pophale2023. The pretrained VGG19 network used in this study is included in Tensorflow 2.0, available at tensorflow.org.

## Human research participants

Policy information about studies involving human research participants and Sex and Gender in Research.

| | |
|---|---|
| Reporting on sex and gender | n/a |
| Population characteristics | n/a |
| Recruitment | n/a |
| Ethics oversight | n/a |

Note that full information on the approval of the study protocol must also be provided in the manuscript.

# Field-specific reporting

Please select the one below that is the best fit for your research. If you are not sure, read the appropriate sections before making your selection.

☒ Life sciences ☐ Behavioural & social sciences ☐ Ecological, evolutionary & environmental sciences

For a reference copy of the document with all sections, see nature.com/documents/nr-reporting-summary-flat.pdf

# Life sciences study design

All studies must disclose on these points even when the disclosure is negative.

| | |
|---|---|
| Sample size | No statistical methods were used to predetermine sample sizes. The number of octopus per for behavioral experiments was determined by seeking to match or exceed that of previous publications on cephalopod resting behavior (refs. 6, 16,17). For electrophysiological experiments we sought to maximize the number of recorded animals within the season of octopus availability (wild caught animals). |
| Data exclusions | Animals were excluded from further behavioral analysis if they exhibited signs of stress after several days of being moved into experimental filming tanks. |
| Replication | Experiments reported in this study followed pilot experiments establishing methods for octopus behavioral filming and electrophysiology. Once methodology and instrumentation was established, all replication attempts were successful. Experiments were performed independently multiple times, sample sizes are stated throughout the manuscript. |
| Randomization | The experiments in our study were mostly observational in nature, and did not involve experimental groups. Manipulation experiments used the same animal before/after manipulation as controls. Randomization was thus not relevant to our study. |
| Blinding | Experiments and data analysis were not performed blind to experimental conditions. Most experiments in our study consist of analysis of a biological system without experimental and control groups. Manipulation experiments largely relied on automated analysis comparing times before and after experimental treatment. Active bouts were identified manually using a quantitative readout of skin brightness and video confirmation. In all cases blinding was not relevant to our study. |

# Reporting for specific materials, systems and methods

We require information from authors about some types of materials, experimental systems and methods used in many studies. Here, indicate whether each material, system or method listed is relevant to your study. If you are not sure if a list item applies to your research, read the appropriate section before selecting a response.

## Materials & experimental systems

| n/a | Involved in the study |
|-----|----------------------|
| ☒ | ☐ Antibodies |
| ☒ | ☐ Eukaryotic cell lines |
| ☒ | ☐ Palaeontology and archaeology |
| ☐ | ☒ Animals and other organisms |
| ☒ | ☐ Clinical data |
| ☒ | ☐ Dual use research of concern |

## Methods

| n/a | Involved in the study |
|-----|----------------------|
| ☒ | ☐ ChIP-seq |
| ☒ | ☐ Flow cytometry |
| ☒ | ☐ MRI-based neuroimaging |

## Animals and other research organisms

Policy information about studies involving animals; ARRIVE guidelines recommended for reporting animal research, and Sex and Gender in Research

| Laboratory animals | This study did not involve laboratory animals. |
|---|---|
| Wild animals | Octopus laqueus of unknown sex and age were captured from tidal pools in Okinawa, Japan. Animals were transported to the OIST marine station/main campus in transport tanks. Transport took ~15 minutes. Following behavioral observation, animals were either used in terminal experiments and euthanized under deep anesthesia before brain extraction and analysis, or died following natural senescence. Animals were not released back to the wild. |
| Reporting on sex | Our study used both male and female animals (16 females, 13 males). Active sleep behavior was observed in all octopuses, motivating us to pool data across sexes. Sex was not considered further in the study design. Sex was determined through examination of arms (presence of hectocotylus in males) and/or the presence of eggs in the abdomen of females. |
| Field-collected samples | This study did not involve samples collected from the field |
| Ethics oversight | The experiments in this study were approved by the OIST Animal Care and Use Committee under approval numbers 2019-244-6 and 2022-364. |

Note that full information on the approval of the study protocol must also be provided in the manuscript.

