## [Peer Review File · Nature]

Manuscript Title: Wake-like skin patterning and neural activity during octopus sleep

Reviewer Comments & Author Rebuttals

Reviewer Reports on the Initial Version:

Referees' comments:

Referee #1 (Remarks to the Author):

My review concentrates on behavior, skin patterning and the general impact of the study given my experience with cephalopods. I cannot properly review electrophys or statistics.

Using a nocturnal species is a bit different but OK since it complements previous studies on octopus sleep.

Overall the behavioral methods appear to be adequate. That is, the lighting levels and periods evoked fairly convincing wake/sleep cycles, and the filming methods were appropriate (but see comment below). Overall, the section on Behavioral signatures of sleep (Line 83 section) is convincing and the summary sentence Line 116 is supported by data in Fig 1.

It would be useful to have some words devoted to the lux readings for diurnal/nocturnal periods. For example, Line 324 indicates day lighting at 654 lux and red night lighting at 129 lux. Put some perspective here about how that relates to the lux these intertidal octopuses might encounter in nature (both day and night). The lights are elsewhere listed as having brightness of 3,000 lux for animal movements Line 330) and 3,933 lux (Line 339) for Canon high res photos. What effects might these variable light intensities have had on your sleep/wake results?

Fig 4d: patterns 1,2,3 do correspond in general to the camouflage pattern designs called Mottle, Uniform, Disruptive, respectively, from literature on octopuses and cuttlefish ... The summary sentence Line 248 is factual and helpful in the present analysis, but providing context for this pattern from known camouflage patterns of octopuses would also be useful and appropriate. For example, the Hanlon lab in Woods Hole has published several octopus camouflage papers and ethograms if you decide to broaden the context of your selective analysis on mantle-only groupings of chromatophores.

DISCUSSION

The Discussion is disappointing - I am a strong proponent of concise writing - but this discussion is ultra-short and lacks a critical appraisal (i.e., weak as well as strong aspects of the current paper) ... or competing hypotheses of function, etc.

Line 281. This sentence is about "cephalopod" skin patterns so you need to give credit to

someone besides your self (citation 36) for this. There are numerous Packard, Boycott, Hanlon and Messenger papers on the hierarchy of body patterns (both in the brain and skin) and you need to reference at least 2-3 of them to be fair .. and for future researchers to be aware of those papers when they read yours.

Some general summary thoughts:

The sleep deprivation experiments are important to this type of study, and the information on gross brain anatomy and its general relationship to electrophys results collectively help clarify what may be happening to octopuses on a daily basis.

However, you have missed an important paper that must be cited - to compare your brain atlas finding (especially the vertical lobe) to the many reported in this 2022 paper:

Chung, W. S., Kurniawan, N. D., & Marshall, N. J. (2022). Comparative brain structure and visual processing in octopus from different habitats. *Current Biology*, 32(1), 97-+. doi:10.1016/j.cub.2021.10.070

Your choice of species is a bit unorthodox ... there is practically no literature on *O. laqueus* (compared to *O. cyanea*, *O. vulgaris* and many others) so it is difficult to make too many generalizations (which is a negative factor for papers considered for journals like Nature).

Importantly, do you know anything about natural patterning of this species' skin? An ethogram of body patterning for this species (or any species used for sleep interpretations) would provide a more robust and realistic baseline for comparing skin patterning.

A seemingly major omission is this: why did you not measure Eye Movements for REM. There were plenty of cameras ... but they seem mostly to be aimed from top down. Simple GoPro cameras positioned horizontally would have been a way to record eye movements. The one supplementary video (side view) shows eye movements ... and the covering of the eye and pupil all have action. This omission is perplexing to readers/reviewers, and distracts seriously from an otherwise very interesting endeavor.

Respectfully submitted, Roger Hanlon

Referee #3 (Remarks to the Author):

Pophale et al. describe a technically innovative approach to determining whether Octopuses have a bona-fide active sleep state akin to vertebrate REM as well as oscillation activities reminiscent of NREM spindles. The study not only finds that many behavioral sleep criteria are satisfied by *O. laqueus*--similar to that of other cephalopods--but also shows convincing evidence bona-fide multi-stage sleep based on assaying sleep deprivation rebound, neural signatures and sleep patterning. What's particularly impressive is the neural activity recording approach and its registration to brain

structure, skin patterning, and behavior.

This study will be an important and exciting contribution to the literature, adding to the growing understanding that sleep is a universal animal trait with diverse neuronal states irrespective of CNS architecture.

Some points to consider before being fully ready for publication:

Major:

1. This paper has potential to be the landmark study of Cephalodian sleep. In this context, it is important for completeness that there is a more thorough presentation of the sleep architecture of *O. Lapeus*. In a typical 24h period, how many wake, QS, AS bouts are there and what are their lengths and distribution in terms of a potential ultradian rhythm/cycle? If not mistaken, this ought to be derivable from the existing behavioral recordings.

2. The authors have done an excellent behavioral assessment of sleep, but a description of muscle tone across the sleep/wake periods is a glaring omission. Muscle tone and myotonic twitches are critical parameters for REM and NREM scoring. Is it technically possible to record muscle activity with Neuropixels? Is there an argument to be made that the motor control of skin pigmentation or papillae extension is an accurate proxy for muscle tone? If so, some evidence of the loss of muscle tone specifically during AS would be a key clinching finding, particularly for readers coming from a polysomnographic tradition.

Other physiological parameters that, while presented in the study, require further attention and detail are eye movements and cardio-respiratory output. Can a video analysis of (ar)rhythmicity of mantle movement (proxy for respiration) be performed across wake, QS and AS to further disambiguate QS from AS/wake? Achieving such additional detail would yield a gold-standard description that would be extremely convincing.

3. Some clarification is sought for some of the choices made to define the different sleep states. What is the rationale for the QS data to be sampled with a filter of 4-40hz considering the significance of the delta band in amniotic deep sleep/SWS? What was the rationale for using a cutoff of 10 second as "noise" for calculating AS? Related to this, how did the authors determine that the brief coloration and movements seen in QS was not a very brief AS or even a brief wake state? The latter two possibilities would change the sleep architecture assumptions somewhat.

4. Since skin pigmentation appears to be a robust cue of AS, could the sleep deprivation of AS specifically be performed to show sleep-stage specific homeostasis and thereby strengthen the case for a bona-fide AS state?

5. There is a lack of precision in the description of cited work and contextualizing the study. For example, the Meisel lab has already established, albeit in different species, behavioral sleep, multi-stage QS and AS, adopting techniques to examine arousal threshold and skin pigmentation characteristics of sleep in Octopus. This is not the impression the reader gets when reading the

manuscript. Please make sure it is made clear in your description and discussion which results presented are consistent with previous work and which are novel.

6. How did the authors disambiguate a contribution of anesthetic effects to the neural dynamics recorded?

Minor:

1. Reference to Irene Tobler re: behavioral sleep criteria would be appropriate. Reduce the number of references to cognitive function of sleep (not relevant/assayed in this study) and move them to the discussion.

2. Any speculation of the potential function of skin pigmentation during AS for discussion purposes?

3. Why was *O. Lapeus* chosen above *O. Vulgaris/Insularis*? Does it have any advantages the authors could describe to the reader?

4. Could the authors spell out the significance of their temperature reduction assay?

5. Please replace "rest" by "sleep".

6. "Octopuses possess 2 stages of sleep:". Please replace by "at least 2 stages of sleep".

7. zscored -> zscored (Methods > Behavioral analysis > Color flashes during QS paragraph > line 7). Please number pages and lines for subsequent rounds of review.

8. Include an awake behavior movie so reader can readily compare with Supplementary Movie 1.

9. Please annotate the movies - e.g. a scale bar for Supplementary Movie 1.

Referee #4 (Remarks to the Author):

This study identifies three behavioural states in an octopus, which are defined by the strength of mechanical stimulus needed to arouse the animal. These are named 'awake', 'quiet sleep' and 'active sleep'; the latter requires the strongest stimulus to arouse the animal. Active sleep last about 1-minute bouts occur hourly during quiet sleep, and are associated with disturbances to the skin pattern and body movement. Temporary deprivation of active sleep results in a compensatory increase in the following days, suggesting that it is required and under homeostatic control.

It is further shown that sleep follows a circadian rhythm (Octopus is nocturnal). Electrical activity in the brain during active sleep is greater than during quiet sleep, and resembles that when the animal is awake.

The final section looks at the animals' coloration patterns, and how they change over time. A clever and innovative analysis shows that during active sleep the octopus displays a range of patterns that are also expressed when the animal is awake, and that these are expressed in no particular sequence - such behaviour might be construed as analogous to dreaming, which perhaps explains the choice of title

Overall this is an impressive study. It is a nicely written well structured account of the two sleep-like states, based on four different methods (arousal, homeostasis, neural activity and skin patterning). The similarities to vertebrate REM and slow wave sleep states are clear, and imply functional convergence.

My chief reservations are due to the novelty and ambition of the work, rather than fundamental weaknesses. Recordings of neural activity (local field potential), skin patterning and new methods (at least in octopus) they are data-rich and complex, so although the conclusions reached seem fine, the non-specialist might have some difficulty in understanding the findings and their interpretation. In particular that the body patterning is 60-dimensional, which is high for example compared to the number of pattern components. One wonders if the results would be been similar if a lower dimensional representation was used - say the first 6 principal components.

There is work on sleep like states including REM sleep in arthropods e.g. Rößler, Daniela C., et al." PNAS 119.33 (2022): e2204754119, and references therein.

Daniel Osorio

Referee #5 (Remarks to the Author):

in this manuscript, the authors set out to clearly define the phenomenon of active sleep in the octopus, which conveys many similarities to awake behaviour (skin patterning and neural activity) while being distinctly different from quiet sleep, a more prevalent form of sleep in the octopus. This is a really amazing paper. First the topic of two modes of sleep in the octopus- a highly intelligent animal while evolutionarily so different from mammals + birds (that have their own form of "active sleep") is fascinating. Secondly, to perform neuropixel recordings in an octopus is a technical tour-de-force.

There are a number of issues that revolve around having a clearer and more statistically robust demonstration that two things are similar (e.g. awake vs active sleep). That being said, I still remain very impressed with this work.

Main concerns

1. As this is an invertebrate, how do you prevent movement of the probe in your recordings. And if there is any movement, how do you make sure LFP during active sleep is not affected by EMG or noise. Can you show data from the octopus in the awake state both during movement and not during movement, to demonstrate similar brain activity?

2. For Fig 2 (c-j) you should include quiet sleep as a reference. It is as important to see that active sleep is different from quiet sleep in brain activity as it is similar to the awake state.

3. Fig 4-It is interesting to see the skin patterning that happens during active sleep, but I am a bit unclear what you are trying to show. First to show a match (either across octopus or between awake and active sleep) it is important to statistically demonstrate that the match of the patterns would not happen by chance, accounting for multiple comparisons if you have multiple templates. One possibility that could help is to expose the octopus to a new environment to evoke new camouflage patterns. Skin patterns during sleep should differ before and after this exposure. Also if two octopus are having the same patterns (4c), this seems to detract from the significance of 4d and the statement in the discussion that this may be similar to replay in rodents and birds. The fact that the trajectories are different is expected by chance. The fact that there is a distinct set of visual patterns that an octopus cycles through may simply be an epiphenomenon of how skin patterning is controlled by the brain, and not a byproduct of awake behaviour.

Minor comments

Fig 1g- while it is helpful to see the z-scored movement, it would also be helpful to see the non-normalised version (perhaps as a supplementary figure), so we can see how the baseline activity changes between states.

Because you are recording sparsely brain regions across animals (which is understandable), it is worthwhile to show reliability of your signal in one brain region (sFL?) which was targeted at a similar location in multiple animals.

What is the threshold for active sleep in extended data figure 2c (draw threshold)

If you are running parametric tests, you should demonstrate that the distribution is normal.

Extended data figure 5- there seems to be a qualitative difference in skin brightness between head fixed and freely behaving (e.g. refractory period, multiple peaks). How is the AS duration calculating, and would changing this threshold alter the results. Essentially it would be helpful to further demonstrate the robustness.

Why does the LFP analysis stop at 40 Hz? Why not continue to higher frequencies and MUA activity?

Fig 1c y-axis should be minutes

Author Rebuttals to Initial Comments:

We thank the reviewers for the very constructive feedback. We reply below in black to the reviewer comments in blue.

Referee #1 (Remarks to the Author):

My review concentrates on behavior, skin patterning and the general impact of the study given my experience with cephalopods. I cannot properly review electrophys or statistics.

Using a nocturnal species is a bit different but OK since it complements previous studies on octopus sleep.

We have now added some additional justification of our use of *O. laqueus* in our methods. This paragraph reads (line 388):

“O. laqueus were carefully selected for this study after assessing multiple other options due to 1) Their compact brain and body size made them suitable for Neuropixels recording and light sheet imaging 2) Their white resting skin pattern aided detection of AS bouts 3) Their nocturnal behaviour means we could film sleep behaviour under white lighting and 4) they were locally available, a regulatory requirement for keeping in the OIST marine station. ...”

Overall the behavioral methods appear to be adequate. That is, the lighting levels and periods evoked fairly convincing wake/sleep cycles, and the filming methods were appropriate (but see comment below). Overall, the section on Behavioral signatures of sleep (Line 83 section) is convincing and the summary sentence Line 116 is supported by data in Fig 1.

Thank you.

It would be useful to have some words devoted to the lux readings for diurnal/nocturnal periods. For example, Line 324 indicates day lighting at 654 lux and red night lighting at 129 lux. Put some perspective here about how that relates to the lux these intertidal octopuses might encounter in nature (both day and night). The lights are elsewhere listed as having brightness of 3,000 lux for animal movements (Line 330) and 3,933 lux (Line 339) for Canon high res photos. What effects might these variable light intensities have had on your sleep/wake results?

We have now added some text in our methods to help put the experimental light levels in context, and to report what we saw differ across light levels. In Extended Data Fig. 5 we compare our electrophysiological experiments (1034 lx) and our low resolution behavioural experiments (654/129 lx). They show similar AS durations and intervals. This paragraph reads (line 435):

“To put lighting levels in context, daylight ranges from 10-25 klx when sunny, ~1000 lx when overcast. Nighttime light levels range from ~0.3 lx during a full moon to ~0.002 lx when the moon is not visible. Animal sleep time, AS duration, and interval appeared normal under various experimental lighting conditions (Extended Data Fig. 5). Darker skin patterns did not appear under 3 klx lighting while awake, but were observed under 1 klx lighting.”

Fig 4d: patterns 1,2,3 do correspond in general to the camouflage pattern designs called Mottle, Uniform, Disruptive, respectively, from literature on octopuses and cuttlefish ... The summary sentence Line 248 is factual and helpful in the present analysis, but providing context for this pattern from known camouflage patterns of octopuses would also be useful and appropriate. For example, the Hanlon lab in Woods Hole has published several octopus camouflage papers and ethograms if you decide to broaden the context of your selective analysis on mantle-only groupings of chromatophores.

We agree, and have added this point to our discussion, citing octopus camouflage/ethogram work of the Hanlon lab and others. The paragraph reads (line 336):

"...Full investigation of the function of AS will require studying whether patterns can be manipulated, as well as a greater understanding of the ethological context in which waking skin patterns are expressed⁵³⁻⁵⁷."

To further and deepen the ethological context of our skin pattern analysis we now include Supplementary Video 4, which includes footage of *O. Laqueus* adopting similar patterns in the field and during AS. We also add an image collection of *O.Laqueus* in their natural environments, writing (line 289)

"While waking, O. laqueus can generate a range of skin patterns to camouflage in different natural environments (Extended Data Fig. 9), as well as for social and threat displays. ..."

DISCUSSION

The Discussion is disappointing - I am a strong proponent of concise writing - but this discussion is ultra-short and lacks a critical appraisal (i.e., weak as well as strong aspects of the current paper) ... or competing hypotheses of function, etc.

We have extended our discussion, adding points on 1) the possible adaptive significance of AS skin patterning behaviour, 2) what our paper does not show or prove (mechanistic understanding, functional relevance) 3) the critical need for ethological studies of octopus waking behaviour for evaluating hypotheses of function, and 4) the general significance of observing 2-stage sleep so broadly across animals.

Line 281. This sentence is about “cephalopod” skin patterns so you need to give credit to someone besides your self (citation 36) for this. There are numerous Packard, Boycott, Hanlon and Messenger papers on the hierarchy of body patterns (both in the brain and skin) and you need to reference at least 2-3 of them to be fair .. and for future researchers to be aware of those papers when they read yours.

You are quite right, we apologise for the oversight. We have added additional references from Packard, Boycott, Hanlon, and Messenger.

Some general summary thoughts:

The sleep deprivation experiments are important to this type of study, and the information on gross brain anatomy and its general relationship to electrophys results collectively help clarify what may be happening to octopuses on a daily basis.

However, you have missed an important paper that must be cited - to compare your brain atlas finding (especially the vertical lobe) to the many reported in this 2022 paper:

Chung, W. S., Kurniawan, N. D., & Marshall, N. J. (2022). Comparative brain structure and visual processing in octopus from different habitats. *Current Biology*, 32(1), 97-+. doi:10.1016/j.cub.2021.10.070

We have now added this reference and given some comparison with the brain of *O. laqueus* and those profiled in this study, writing in our methods section (line 392):

“...The brain of O laqueus resembles coastal diurnal octopuses in possessing a 7-gyrus vertical lobe (Extended Data Fig. 3d). The vertical lobe occupies 9.05% of central brain volume, slightly higher than other coastal nocturnal species, like the commonly studied O. vulgaris and O. bimaculoides⁶⁴.”

Your choice of species is a bit unorthodox ... there is practically no literature on *O. laqueus* (compared to *O. cyanea*, *O. vulgaris* and many others) so it is difficult to make too many generalizations (which is a negative factor for papers considered for journals like Nature).

Importantly, do you know anything about natural patterning of this species' skin? An ethogram of body patterning for this species (or any species used for sleep interpretations) would provide a more robust and realistic baseline for comparing skin patterning.

As you say, there unfortunately hasn't been a thorough ethological study of this species. While we can associate sleeping and waking patterns, the ethological situations where many of these patterns are expressed remain to be explored, and indeed may be central to interpreting the content of AS skin patterning. We have added a discussion of this point to highlight the limitation of conclusions and the need for further studies, writing (line 336):

"... Full investigation of the function of AS will require studying whether patterns can be manipulated, as well as a greater understanding of the ethological context in which waking skin patterns are expressed⁶³⁻⁵⁷."

Thanks to your encouragement to look further into the literature on this topic we have actually found one mention of waking skin pattern behaviour where they study dynamics skin patterns (How et al. 2017). They describe a dark pulse of chromatophore expansion seen during foraging. We also observe this behaviour, especially in the field, and it appears during active sleep. We have added a video (Supplementary Video 4) showing the chromatic pulse during wake and sleep, and cite the How et al. study. To add some further context to our skin pattern analysis we have also collected some field images that we refer to in our text and add as Extended Data Fig. 9 (shown above).

A seemingly major omission is this: why did you not measure Eye Movements for REM. There were plenty of cameras ... but they seem mostly to be aimed from top down. Simple GoPro cameras positioned horizontally would have been a way to record eye movements. The one supplementary video (side view) shows eye movements ... and the covering of the eye and pupil all have action. This omission is perplexing to readers/reviewers, and distracts seriously from an otherwise very interesting endeavor.

Thank you for pointing this out. We have now analysed additional high resolution video shot from a side-view, tracking eye movements, as well as body twitching, breathing rate, and breathing variability. All show reliably substantial increases, similar to what you can see in Supplementary video 1. We have added this to the description of active sleep in Figure 1 (left, below) and Extended Data Fig. 1c (right, below).

Respectfully submitted, Roger Hanlon

Thank you Dr. Hanlon for your very helpful review.

Referee #3 (Remarks to the Author):

Pophale et al. describe a technically innovative approach to determining whether Octopuses have a bona-fide active sleep state akin to vertebrate REM as well as oscillation activities reminiscent of NREM spindles. The study not only finds that many behavioral sleep criteria are satisfied by *O. Lapeus*--similar to that of other cephalopods--but also shows convincing evidence bona-fide multi-stage sleep based on assaying sleep deprivation rebound, neural signatures and sleep patterning. What's particularly impressive is the neural activity recording approach and its registration to brain structure, skin patterning, and behavior.

This study will be an important and exciting contribution to the literature, adding to the growing understanding that sleep is a universal animal trait with diverse neuronal states irrespective of CNS architecture.

Thank you very much.

Some points to consider before being fully ready for publication:

Major:

1. This paper has potential to be the landmark study of Cephalodian sleep. In this context, it is important for completeness that there is a more thorough presentation of the sleep architecture of *O. Lapeus*. In a typical 24h period, how many wake, QS, AS bouts are there and what are their lengths and distribution in terms of a potential ultradian rhythm/cycle? If not mistaken, this ought to be derivable from the existing behavioral recordings.

We have now calculated this, writing in our discussion of Fig. 1d,e (line 114):

"... The rate of active bouts was strongly modulated over 24 hours, peaking during the 12 hours of subjective daytime. In a typical 24 hour period at 22°C animals underwent 10 ± 3.5 active bouts of $75 \text{ s} \pm 28 \text{ s}$ in duration and 12 ± 3 QS bouts of $50.5 \text{ min} \pm 16.43 \text{ min}$ in duration ($N=3$ animals, $\text{mean} \pm \text{SD}$). This modulation persisted through prolonged periods of constant light or darkness (Fig. 1i,j), suggesting internal control²⁷ (Rayleigh-test, lights on: $p=1.5e-12$, $N=322$ bouts, lights off: $p=3.0e-13$, $N=318$ bouts)."

2. The authors have done an excellent behavioral assessment of sleep, but a description of muscle tone across the sleep/wake periods is a glaring omission. Muscle tone and myotonic twitches are critical parameters for REM and NREM scoring. Is it technically possible to record muscle activity with Neuropixels? Is there an argument to be made that the motor control of skin pigmentation or papillae extension is an accurate proxy for muscle tone? If so, some evidence of the loss of muscle tone specifically during AS would be a key clinching finding, particularly for readers coming from a polysomnographic tradition.

This is an interesting suggestion. We don't think it is technically feasible to record muscle tone using Neuropixels probes. In the brain we did everything possible to stabilise things in order to record neural activity uncontaminated by movement artefacts. We now show a schematic of our head fixation technique (left, Extended Data Fig. 6a), and an example of our recording stability during waking movement (right, Extended Data Fig. 6c), showing a period of time around a sleep-wake transition (black arrow). Neural activity in the sFL (black, top) increases and mantle coloration (red) darkens upon waking. Activity in the iFL (black, bottom), remains quiet. There are two periods of transient large movements, which are not prominent in either LFP recording.

The octopus body is so flexible that it would almost definitely break probes placed outside our head fixation. Chromatophore expansion and papillae extension operate under muscular control, and we additionally observe increased arm, body, and head twitches during AS. We now profile profile twitching body movements in Figure 1 and Extended Data Fig. 1:

We don't see coordinated movements during AS, possibly indicating some muscles are inhibited while others are clearly activated during AS. We have added this point to our discussion, writing (line 321):

“... Active sleep resembles vertebrate REM sleep in terms of wake-like neural activity accompanying eye and body twitches^{17,41}. Coordinated postural changes (e.g. arm reaching) are not seen during AS, potentially indicating some level of muscular inhibition. However, a lack of anatomical homology complicates comparison with the atonia of skeletal muscles found in vertebrate REM sleep⁴¹. ...”

Other physiological parameters that, while presented in the study, require further attention and detail are eye movements and cardio-respiratory output. Can a video analysis of (ar)rhythmicity of mantle movement (proxy for respiration) be performed across wake, QS and AS to further disambiguate QS from AS/wake? Achieving such additional detail would yield a gold-standard description that would be extremely convincing.

This is a great suggestion, and indeed it can. We now used optic flow to track mantle movement during wake, QS, and AS, and show increased rhythmicity (measured as the coefficient of variation of breathing rate) during QS relative to AS/wake. This is now Extended Data Fig. 1b (left, below). We have also added an AS/QS comparison of breathing rate to Figure 1e (right, below).

3. Some clarification is sought for some of the choices made to define the different sleep states. What is the rationale for the QS data to be sampled with a filter of 4-40hz considering the significance of the delta band in amniotic deep sleep/SWS?

We have now added an analysis across all brain regions and recordings of both low frequency (0.1-10 Hz) and higher frequency (20-150 Hz) LFP frequencies during QS, AS and wakefulness (Fig. 3i,j). These data better support our finding of no prominent delta band activity in the regions of the octopus central brain we recorded from. In Fig. 4 We focus on a 4-40 Hz LFP band for the analysis of the spindle-like oscillations we found during QS, which had power at 12-18 Hz (Fig. 4c).

What was the rationale for using a cutoff of 10 second as "noise" for calculating AS?

We have now improved and simplified our AS duration calculation, and do not have any cutoff time. The technique is now explained in the Methods section (line 600):

"... AS duration was determined by considering a window 10s before:100s after AS start times. These time series were z-scored, and low pass filtered at 0.1 Hz using a 2-pole Butterworth filter. The length of the largest continuous stretch of data falling below a threshold was taken as the AS duration. We explored a range of thresholds (Extended Data Fig. 5d), deciding on 0.2 as a good subjective match to video data."

In addition to our parameter sweep (left, below), we have added examples of data filtering and detected start and end times, for both head fixed and freely behaving animals, to Extended Data Fig. 5 (right, below).

Related to this, how did the authors determine that the brief coloration and movements seen in QS was not a very brief AS or even a brief wake state? The latter two possibilities would change the sleep architecture assumptions somewhat.

We currently do not know whether the brief coloration events during QS represent very brief AS, awakenings, a part of QS, or a distinct sleep stage. They differ from AS/waking in their much shorter duration, smaller level of neural activity, and lack of pronounced movements, but the correlation between neural activity during waking, AS and QS brief colorations is very high. We now add a discussion of this point, which is well taken. We feel that the reviewer's suggestion to change our conclusion to 'octopuses possess at least 2 stages of sleep' is the most appropriate stance to take. The paragraph reads (line 318):

"...The brief flashes of coloration seen during quiet sleep are accompanied by neural activity levels resembling waking, albeit at lower amplitude. Whether this constitutes brief periods of waking, micro-arousal states⁴⁰, a kind of QS, or a distinct sleep stage remains unclear. ..."

4. Since skin pigmentation appears to be a robust cue of AS, could the sleep deprivation of AS specifically be performed to show sleep-stage specific homeostasis and thereby strengthen the case for a bona-fide AS state?

Thank you for proposing this experiment. In carrying it out we were surprised to find a remarkable sensitivity to AS homeostasis. We selectively interrupted a subset of AS bouts, and found that the amount of quiet sleep before the next AS bout was much shorter following interrupted bouts than when uninterrupted. We now schematize and report this result in Fig. 2c and d.

5. There is a lack of precision in the description of cited work and contextualizing the study. For example, the Meisel lab has already established, albeit in different species, behavioral sleep, multi-stage QS and AS, adopting techniques to examine arousal threshold and skin pigmentation characteristics of sleep in Octopus. This is not the impression the reader gets when reading the manuscript. Please make sure it is made clear in your description and discussion which results presented are consistent with previous work and which are novel.

We apologise for the lack of clarity. We now write the following, citing Meisel et al. and Medeiros et al. in reference to octopus sleep work, and Irene Tobler in reference to behavioural sleep criteria (line 87):

“... Sleeping cephalopods¹⁹ have been observed to undergo rhythmic bouts of body twitches and rapid changes in skin patterning^{6,20}, mediated by neural control of large populations of skin pigment cells (chromatophores)²¹ among other specialised cell types²². In octopus, this has been termed ‘active sleep’ (AS), and is accompanied by an increased arousal threshold, one of several criteria of sleep^{17,23,24}. Expanding on this previous work, we tested whether octopuses possess two stages of sleep behaviour. We then examined neural activity and skin pattern dynamics during sleeping and waking, by developing novel methods for behavioural recording and quantification, light sheet imaging, and LFP recordings using Neuropixels probes in these soft bodied animals.”

6. How did the authors disambiguate a contribution of anesthetic effects to the neural dynamics recorded?

Thank you, this is an important point. Octopuses rapidly recovered from ethanol anaesthesia, demonstrating seemingly normal behaviour minutes after being transferred to normal sea water. Additionally, animals began AS 12.7 ± 6.6 (SD) hours after recording

began, decreasing the likelihood of anaesthesia being an influencing factor. We now state these points in our Methods section (lines 491 and 510).

Interested in knowing if the experimental conditions of our neural recordings (including previous anaesthesia, head fixation, and probe implantation) influenced sleep behaviour, we compared it with normally sleeping animals. We found that the rhythmic alternation between AS/QS, AS duration, and AS interval were similar in both cases. We show this data in Extended Data Fig. 5, and now refer to it in our methods section.

Minor:

1. Reference to Irene Tobler re: behavioral sleep criteria would be appropriate. Reduce the number of references to cognitive function of sleep (not relevant/assayed in this study) and move them to the discussion.

We have added a reference to Irene Tobler on behavioural sleep criteria, and moved some references to sleep function to the discussion.

2. Any speculation of the potential function of skin pigmentation during AS for discussion purposes?

We now elaborate on potential functions of AS skin patterning in the discussion. The paragraph reads (line 328):

“Additionally, during active sleep octopuses rapidly transition through sets of skin patterns which strongly resemble those seen while awake. This normally occurs in the safety of the octopus den, and therefore does not regularly broadcast the animal’s position to predators. Why do octopuses perform this dramatic sleep behaviour? One possibility is that it represents periods of offline refinement of skin pattern control, analogous to processes thought to occur during vertebrate motor learning^{42–44}. Another possibility is that it reflects the

reactivation of neural activity underlying waking experience more broadly, reminiscent of vertebrate phenomena linked to memory consolidation such as rodent hippocampal replay⁴⁵⁻⁴⁹ and the structured activity in the head direction system during mammalian REM sleep⁵⁰⁻⁵². Full investigation of the function of AS will require studying whether patterns can be manipulated, as well as a greater understanding of the ethological context in which waking skin patterns are expressed⁵³⁻⁵⁷.

3. Why was *O. Lapeus* chosen above *O. Vulgaris/Insularis*? Does it have any advantages the authors could describe to the reader?

We have now added some additional justification of our use of *O. laqueus* in our methods section, writing (line 388):

“O. laqueus were carefully selected for this study after assessing multiple other options due to 1) Their compact brain and body size made them suitable for Neuropixels recording and light sheet imaging 2) Their white resting skin pattern aided detection of AS bouts 3) Their nocturnal behaviour means we could film sleep behaviour under white lighting and 4) they were locally available, a regulatory requirement for keeping in the OIST marine station. ...”

4. Could the authors spell out the significance of their temperature reduction assay?

We now write in our discussion, citing literature on temperature robustness of neural systems (line 312):

“... Rhythmic AS bouts are homeostatically regulated and robust to temperature and lighting manipulations, indicative of an actively maintained biological phenomenon of central importance^{37,38}. ...”

5. Please replace “rest” by “sleep”.

We of course agree with this sentiment. We have removed ‘rest’ from the manuscript except for mentioning a ‘resting posture’. Part of this study concerns establishing that active bouts constitute a stage of sleep in this species, and we thought it would be awkward to call it sleep before we show it is sleep. We therefore now refer to ‘active bouts’ until we present our sleep criteria experiments, and then afterwards refer to ‘active sleep’.

6. “Octopuses possess 2 stages of sleep.” Please replace by “at least 2 stages of sleep”.

Excellent suggestion, we have replaced.

7. zscored -> zscored (Methods > Behavioral analysis > Color flashes during QS paragraph > line 7). Please number pages and lines for subsequent rounds of review.

We apologise, and have fixed this. In our uploaded manuscript copy we saw line numbers, added by the Nature system. We have uploaded our latest manuscript version as a pdf with line and page numbers to avoid any issues this round.

8. Include an awake behavior movie so reader can readily compare with Supplementary Movie 1.

We now include Supplementary Video 2, a video shot from the side as in Supplementary Video 1, showing awake behaviour. We also include Supplementary Video 4, which shows examples of matching skin patterning during wake and AS.

9. Please annotate the movies - e.g. a scale bar for Supplementary Movie 1.

We have added scale bars to all Supplementary Videos.

Referee #4 (Remarks to the Author):

This study identifies three behavioural states in an octopus, which are defined by the strength of mechanical stimulus needed to arouse the animal. These are named 'awake', 'quiet sleep' and 'active sleep'; the latter requires the strongest stimulus to arouse the animal. Active sleep last about 1-minute bouts occur hourly during quiet sleep, and are associated with disturbances to the skin pattern and body movement. Temporary deprivation of active sleep results in a compensatory increase in the following days, suggesting that it is required and under homeostatic control.

It is further shown that sleep follows a circadian rhythm (Octopus is nocturnal). Electrical activity in the brain during active sleep is greater than during quiet sleep, and resembles that when the animal is awake.

The final section looks at the animals' coloration patterns, and how they change over time. A clever and innovative analysis shows that during active sleep the octopus displays a range of patterns that are also expressed when the animal is awake, and that these are expressed in no particular sequence - such behaviour might be construed as analogous to dreaming, which perhaps explains the choice of title

Overall this is an impressive study. It is a nicely written well structured account of the two sleep-like states, based on four different methods (arousal, homeostasis, neural activity and skin patterning). The similarities to vertebrate REM and slow wave sleep states are clear, and imply functional convergence.

Thank you Dr. Osorio for the kind words.

My chief reservations are due to the novelty and ambition of the work, rather than fundamental weaknesses. Recordings of neural activity (local field potential), skin patterning and new methods (at least in octopus) they are data-rich and complex, so although the conclusions reached seem fine, the non-specialist might have some difficulty in understanding the findings and their interpretation. In particular that the body patterning is 60-dimensional, which is high for example compared to the number of pattern components. One wonders if the results would be been similar if a lower dimensional representation was used - say the first 6 principal components.

Thank you for this feedback. We have now expanded our discussion section in the effort to better explain our findings and their interpretation. We have also repeated our analysis of skin pattern dynamics using the top 6 PCs, producing similar results. AS skin pattern trajectories occupy a largely shared pattern space (Silhouette score: 0.076 ± 3.148), are diverse and intersect each other. Dynamic time warping does not reduce the inter-trajectory distances to those of the nearest inter-trajectory distances, and waking patterns appear nearby active sleep patterns. As expected for data which are actually ~60 dimensional, all points fall at closer distances to each other when only considering the top 6 PCs. We show these results in Extended Data Fig. 8b.

There is work on sleep like states including REM sleep in arthropods e.g. Rößler, Daniela C., et al." PNAS 119.33 (2022): e2204754119, and references therein.

Thank you for the reference. We have added it in our expanded discussion about the significance of observing 2-stage sleep across multiple animal groups. The paragraph reads (line 347):

“While initially observed in humans², recent work has established 2-stage sleep across many vertebrate species¹⁻⁴. Our results complement several behavioural reports in cephalopods^{6,20} and arthropods⁶² of similar active and quiet sleep stages. Given the evolutionary distances, these phenomena likely evolved independently from each other, and may represent convergent solutions to shared problems facing complex agents⁶³. If such solutions indeed exist, then the high dimensional and interpretable readout of neural activity in octopus active sleep skin patterns may help to uncover general principles of 2-stage sleep.”

Daniel Osorio

Referee #5 (Remarks to the Author):

in this manuscript, the authors set out to clearly define the phenomenon of active sleep in the octopus, which conveys many similarities to awake behaviour (skin patterning and neural activity) while being distinctly different from quiet sleep, a more prevalent form of sleep in the octopus. This is a really amazing paper. First the topic of two modes of sleep in the octopus- a highly intelligent animal while evolutionarily so different from mammals + birds (that have their own form of “active sleep”) is fascinating. Secondly, to perform neuropixel recordings in an octopus is a technical tour-de-force.

There are a number of issues that revolve around having a clearer and more statistically robust demonstration that two things are similar (e.g. awake vs active sleep). That being said, I still remain very impressed with this work.

Thank you very much.

Main concerns

1. As this is an invertebrate, how do you prevent movement of the probe in your recordings. And if there is any movement, how do you make sure LFP during active sleep is not affected by EMG or noise. Can you show data from the octopus in the awake state both during movement and not during movement, to demonstrate similar brain activity?

This is an important point, thank you. We found that the octopus brain is surrounded by a relatively rigid layer of cartilage that we could head fix to. We now include a detailed diagram of our head fixation setup (left, Extended Data Fig. 6a), and we describe our technique in the Methods section. We have also added the requested figure (right, Extended Data Fig. 6b), showing a period of time around a sleep-wake transition (black arrow). Neural activity in the sFL (black, top) increases and mantle coloration (red) darkens upon waking. Activity in the iFL (black, bottom), remains quiet. There are two periods of transient large movements, which are not prominent in either LFP recording. We note that the brain region specificity of our LFP signals (Extended Data Fig. 4), and consistency in signals across multiple hours of recordings (Extended Data Fig 6d, below) also speaks to the stability of our recordings.

2. For Fig 2 (c-j) you should include quiet sleep as a reference. It is as important to see that active sleep is different from quiet sleep in brain activity as it is similar to the awake state.

Thank you for the nice suggestion. We have added quiet sleep as a reference, calculated in the same way as AS/wake, to the violin plots in what is now Fig. 3i and j (left, below). We wanted to keep the 3d brain images with registered Neuropixels probes in the figure but having 6 of them became too crowded. We therefore moved the quiet sleep 3d brain images to Extended Data Fig 7b (right, below).

3. Fig 4-It is interesting to see the skin patterning that happens during active sleep, but I am a bit unclear what you are trying to show. First to show a match (either across octopus or between awake and active sleep) it is important to statistically demonstrate that the match of the patterns would not happen by chance, accounting for multiple comparisons if you have multiple templates.

A priori, the sequence of AS skin patterns could have been stereotyped, octopus specific, and differ from waking patterns. We developed methods for quantifying AS skin patterns, and found they were variable, occupied a shared space, and could be precisely registered with waking patterns. It is worth noting though, that unlike in a situation of a completely random and independent 2d space of pixels, the pattern space is much more constrained.

We agree that given that the patterns are variable and occupy a shared pseudo-constrained space, it is potentially unsurprising that the same patterns appear across bouts and octopuses. We consider this an interesting consequence. Indeed, the biological and ethological constraints on patterning that produce these observations remain to be investigated. We note that here we intentionally opted for a heuristic approach that samples the space, rather than a comprehensive one that fully characterises it quantitatively, which will require analysis much beyond the scope of this paper. We have now clarified in the manuscript the modesty and extent of our claims, and stressed that further investigation of the space of patterns will need to be addressed in future studies. Finally, we have taken to heart the warning about a multiple comparison problem that might arise more strongly in follow up studies that will investigate AS comprehensively, and thank the reviewer for pointing this important issue out.

One possibility that could help is to expose the octopus to a new environment to evoke new camouflage patterns. Skin patterns during sleep should differ before and after this exposure.

We think this is an interesting idea which we plan to test in future work. Skin patterns differing before and after exposure is one of several possible results. Others include the lack of ability to manipulate, longer timescale experience being relevant, or a developmental period where patterns are formed and after which can not be manipulated. We felt it best to split the story here, so as to not detract from our electrophysiology work and sleep criteria results. Describing AS skin patterns and their match to waking patterns sets up this followup study and we think is of general interest.

Also if two octopus are having the same patterns (4c), this seems to detract from the significance of 4d and the statement in the discussion that this may be similar to replay in rodents and birds.

We think it is possible that environmental exposure changes the probability of adopting a pattern, transitioning between certain patterns, changes more subtle spatial structure of patterns, or affects other functions of AS pattern dynamics. Perhaps octopuses show the same patterns similar to how birds sing the same song. We have now expanded our discussion on this topic (please see below).

The fact that the trajectories are different is expected by chance. The fact that there is a distinct set of visual patterns that an octopus cycles through may simply be an epiphenomenon of how skin patterning is controlled by the brain, and not a byproduct of awake behaviour.

We regret not being more clear on this point – we fully agree that this study does not show the function of AS skin patterns, and does not demonstrate a phenomena akin to hippocampal replay. We hope to test functional ideas in followup work. AS skin patterning is interesting, in any case, as it provides a uniquely expressive readout of neural activity in an offline brain. It therefore is potentially useful for understanding motor control of the skin patterning system. We now state these points in an expanded discussion section. It now reads (line 328):

“Additionally, during active sleep octopuses rapidly transition through sets of skin patterns which strongly resemble those seen while awake. This normally occurs in the safety of the octopus den, and therefore does not regularly broadcast the animal’s position to predators. Why do octopuses perform this dramatic sleep behaviour? One possibility is that it represents periods of offline refinement of skin pattern control, analogous to processes thought to occur during vertebrate motor learning^{42–44}. Another possibility is that it reflects the reactivation of neural activity underlying waking experience more broadly, reminiscent of vertebrate phenomena linked to memory consolidation such as rodent hippocampal replay^{45–49} and the structured activity in the head direction system during mammalian REM sleep^{50–52}. Full investigation of the function of AS will require studying whether patterns can be manipulated, as well as a greater understanding of the ethological context in which waking skin patterns are expressed^{53–57}.

Cephalopod skin patterns appear to be organised hierarchically, with putative higher-order motor control circuitry coordinating large groups of chromatophores to generate macroscopic pattern elements^{21,35,58-61}. AS dynamics are consistent with the pseudo-random activation of this high-level control system. It may be possible to infer interactions between motor control elements by studying the statistics of pattern activation. In this way, AS dynamics may also be useful for understanding the logic of waking skin pattern control.”

Minor comments

Fig 1g- while it is helpful to see the z-scored movement, it would also be helpful to see the non-normalised version (perhaps as a supplementary figure), so we can see how the baseline activity changes between states.

We have now implemented a more sensitive measure of movement throughout the manuscript. We report the arousal threshold result without z-scoring in Fig. 2a (left, below), and we include a direct measure of baseline activity in Extended data Fig. 1d (right, below).

Because you are recording sparsely brain regions across animals (which is understandable), it is worthwhile to show reliability of your signal in one brain region (sFL?) which was targeted at a similar location in multiple animals.

Thank you for the suggestion. We now show all recorded AS bouts recorded from 3 animals in the sFL (left), and from the VL (right), demonstrating the reliability of the signal across animals, and difference in signal across brain regions. This is added to Extended Data Fig. 6d.

What is the threshold for active sleep in extended data figure 2c (draw threshold)

Active sleep bouts were identified manually, as dark skin patterning while awake could resemble the mean brightness of an active sleep bout. We have added a description of this to the figure legend (line 953):

“c) Time series of mean skin brightness of three octopuses, simultaneously recorded and automatically segmented using a Mask R-CNN. Blue arrowheads: active rest bouts (manually detected, Methods).”

If you are running parametric tests, you should demonstrate that the distribution is normal.

Thank you for this comment. In all cases our data distributions failed tests of normality, and we therefore moved to running non-parametric tests.

Extended data figure 5- there seems to be a qualitative difference in skin brightness between head fixed and freely behaving (e.g. refractory period, multiple peaks). How is the AS duration calculating, and would changing this threshold alter the results. Essentially it would be helpful to further demonstrate the robustness.

We apologise for the lack of clarity on this point. We have refined and simplified our AS duration calculation, using a threshold crossing of low pass filtered skin brightness data. We describe this in the Methods section, writing (line 600):

“...AS duration was determined by considering a window 10s before:100s after AS start times. These time series were z-scored, and low pass filtered at 0.1 Hz using a 2-pole Butterworth filter. The length of the largest continuous stretch of data falling below a threshold was taken as the AS duration. We explored a range of thresholds (Extended Data Fig. 5d), deciding on 0.2 as a good subjective match to video data.”

We now show examples of head fixed and freely behaving AS bouts, where we mark the start and end time of the bout (left, Extended Data Fig. 5c). We additionally performed a parameter sweep, calculating durations under a range of thresholds for head fixed and freely behaving AS bouts. This shows the robustness of calculating the similarity in AS durations, and where our subjective choice of threshold (0.2) falls (right, Extended Data Fig. 5d).

We agree, it does seem that there are some qualitative differences in skin pattern dynamics and the speed of return to a white pattern after AS bouts. We think this could be due to different levels of sleep depth in the two conditions, or could be a consequence of recording differences (whole body vs mantle). We add these possibilities to the figure legend.

Why does the LFP analysis stop at 40 Hz? Why not continue to higher frequencies and MUA activity?

We stopped at 40 Hz because we didn't see a lot in terms of MUA at higher frequencies in our recordings. Most of the LFP signal came from neuropil regions, where it is possible that neural processes are too fine to pick up unit activity with the 12x12 μm electrodes on Neuropixels probes. We are currently exploring this possibility with imaging approaches. We now have raised our higher frequency LFP band to 20-150 Hz, reporting similar results (please see above). For analysis of spindle-like oscillations we picked 40 Hz because of the lack of power at higher frequencies (Fig. 4c).

Fig 1c y-axis should be minutes

Thank you for catching this.

Reviewer Reports on the First Revision:

Referees' comments:

Referee #3 (Remarks to the Author):

Thank you to the authors for your comprehensive responses. All concerns are addressed.

This reviewer is particularly encouraged to see that AS displays heightened and arrhythmic breathing compared to QS and that this sleep stage is subject to homeostasis, while noting the absence of delta power and the understandable limitations in studying muscle atonia in cephalopods.

Overall, a fantastic addition to the literature!

Referee #4 (Remarks to the Author):

I have nothing to add to my previous general evaluation. This is a very interesting and original study, the revision has addressed the referees' comments.

Referee #5 (Remarks to the Author):

Excellent paper!! The revision has satisfied all my previous remarks, and happy to support its publication at this stage.

-Daniel Bendor